# 3D virtual histopathology of cardiac tissue from Covid-19 patients based on phase-contrast X-ray tomography

Marius Reichardt[1], Patrick Moller Jensen[2], Vedrana Andersen Dahl[2], Anders Bjorholm Dahl[2], Maximilian Ackermann[3], Harshit Shah[4,5], Florian Länger[4,5], Christopher Werlein[4], Mark P Kuehnel[4,5], Danny Jonigk[4,5]*, Tim Salditt[1]*

[1]Institut für Röntgenphysik, Georg-August-Universität Göttingen, Friedrich-Hund-Platz, Göttingen, Germany; [2]Technical University of Denmark, Richard Petersens Plads, Kopenhagen, Denmark; [3]Institute of Anatomy and Cell Biology, University Medical Center of the Johannes Gutenberg-University Mainz, Mainz, Germany; [4]Medizinische Hochschule Hannover (MHH), Hannover, Germany; [5]Deutsches Zentrum für Lungenforschung (DZL), Hannover (BREATH), Hannover, Germany

**\*For correspondence:**
Jonigk.Danny@mh-hannover.
de (DJ);
tsaldit@gwdg.de (TS)

**Competing interest:** The authors declare that no competing interests exist.

**Abstract** For the first time, we have used phase-contrast X-ray tomography to characterize the three-dimensional (3d) structure of cardiac tissue from patients who succumbed to Covid-19. By extending conventional histopathological examination by a third dimension, the delicate pathological changes of the vascular system of severe Covid-19 progressions can be analyzed, fully quantified and compared to other types of viral myocarditis and controls. To this end, cardiac samples with a cross-section of 3.5mm were scanned at a laboratory setup as well as at a parallel beam setup at a synchrotron radiation facility the synchrotron in a parallel beam configuration. The vascular network was segmented by a deep learning architecture suitable for 3d datasets (V-net), trained by sparse manual annotations. Pathological alterations of vessels, concerning the variation of diameters and the amount of small holes, were observed, indicative of elevated occurrence of intussusceptive angiogenesis, also confirmed by high-resolution cone beam X-ray tomography and scanning electron microscopy. Furthermore, we implemented a fully automated analysis of the tissue structure in the form of shape measures based on the structure tensor. The corresponding distributions show that the histopathology of Covid-19 differs from both influenza and typical coxsackie virus myocarditis.

## Editor's evaluation

In this manuscript the authors demonstrate that X-ray imaging delivers more detailed information than standard histology by analyzing 3D information in myocardial tissue obtained from COVID-19 patients. The findings are of particular interest regarding the segmentation of the vascular network and intussusceptive angiogenesis. The authors introduce the utilization of machine learning, and state-of-the-art techniques of X-ray phase contrast which is likely to advance future work in this field. Finally, with this manuscript the authors also provide new, more detailed insights into the pathologies associated with cardiac injury due to COVID-19.

## Introduction

The coronavirus disease 2019 (Covid-19) is caused by the severe acute respiratory syndrome coronavirus (SARS-CoV-2), predominantly entering the body via the respiratory tract. SARS-CoV-2 infects

cells by binding its spike protein to the surface protein angiotensin-converting enzyme 2 (ACE2) of the host cell (*Hoffmann et al., 2020*). Severe cases are most frequently affected by viral pneumonia and acute respiratory distress syndrome (ARDS), with a pathophysiology distinctly different from for example influenza infection (*Ackermann et al., 2020b*). Mediated by a distinct inflammatory microenvironment, an uncontrolled infection can develop and result in massive tissue damage, again primarily reported in the lung. Apart from diffuse alveolar damage, the main histological hallmark of ARDS, specific findings in the lung histopathology are high prevalence of micro-thrombi and high levels of intussusceptive angiogenesis (IA) (*Ackermann et al., 2020a*; *Ackermann et al., 2020b*; *Bois et al., 2021Ackermann et al., 2020c*). The latter is a rapid process of intravascular septation that produces two lumens from a single vessel. It is distinct from sprouting angiogenesis because it has no necessary requirement for cell proliferation, can rapidly expand an existing capillary network, and can maintain organ function during replication (*Mentzer and Konerding, 2014*). The mechanistic link between branch angle remodeling and IA is the intussusceptive pillar. The pillar is a cylindrical 'column' or 'pillar' that is $1\,\mu$m to $3\,\mu$m in diameter (*Ackermann and Konerding, 2015*). In short, the capillary wall extends into the lumen and split a single vessel in two. Opposing capillary walls are first dilated, and intraluminal pillars form at vessel bifurcations by an intraluminal intussusception of myofibroblasts, creating a core between the two new vessels. These cells begin depositing collagen fibers into the core, providing an extracellular matrix (ECM) for the growth of the vessel lumen. The extension of the pillar along the axis of the vessel then results in vessel duplication. These structural changes of the vasculature have been reported in various non-neoplastic and neoplastic diseases (*Erba et al., 2011*; *Albert et al., 2020*, *Ackermann et al., 2012*). These finding underline the notion that Covid-19 is a disease driven by, and centered around, the vasculature with direct endothelial infection, thus providing SARS-CoV-2 an easy entry route into other organs, subsequently resulting in multi-organ damage (*Nishiga et al., 2020*; *Menter et al., 2020*).

Clinically, the heart appears to be a particular organ at risk in Covid-19. Acute cardiac involvement (e.g. lowered ejection fraction, arrhythmia, dyskinesia, elevated cardiac injury markers) is reported in a broad range of cases. In contrast to other respiratory viral diseases affecting the heart (e.g. coxsackie virus), in the few Covid-19 cases reported so far that included cardiac histopathology, no classic lymphocytic myocarditis characterized by a T-lymphocyte predominant infiltrate with cardiomyocyte necrosis was observed (*Gauchotte et al., 2021*; *Kawakami et al., 2021*; *Tavazzi et al., 2020*; *Albert et al., 2020*; *Wenzel et al., 2020*; *Halushka and Vander Heide, 2021*). Furthermore, the underlying pathomechanisms are still poorly understood with both direct virus induced (cellular) damage and indirect injury being discussed (*Zheng et al., 2020*; *Wichmann et al., 2020*; *Gauchotte et al., 2021*; *Chen et al., 2020*; *Deng et al., 2020*; *Zeng et al., 2020*). Particularly, it is not known to which extent the vasculature of the heart, including the smallest capillaries, are affected and whether IA is also a dominant process in this organ. More generally, one would like to delineate the morphological changes of cytoarchitecture from other well described pathologies. Recently, we have used three-dimensional (3d) virtual histology based on phase-contrast X-ray tomography as a new tool for Covid-19 pathohistology and investigated these structural changes in postmortem tissue biopsies from Covid-19 diseased lung tissue using propagation based X-ray tomography (*Eckermann et al., 2020*; *Walsh et al., 2021*). Exploiting phase contrast based on wave propagation, the 3d structure of formalin-fixed, paraffin-embedded (FFPE) tissue–the mainstay for histopathological samples worldwide- can be assessed at high resolution, that is with sub-micron voxel size and with sufficient contrast also for soft and unstained tissues (*Töpperwien et al., 2018*). By relaxing the resolution to voxel sizes in the range of 25 µm and stitching of different tomograms, the entire human organ can be covered and an entire FFPE tissue block 'unlocked' by destruction-free 3d analysis (*Walsh et al., 2021*).

In this work, we now focus on the 3d cytoarchitecture of cardiac tissue. We have scanned unstained, paraffin embedded tissue, prepared by a biopsy punch from paraffin-embedded tissue blocks, collected from patients which have succumbed to Covid-19 (Cov). For comparison, we have scanned tissue from influenza (Inf) and myocarditis (Myo) patients as well as from a control group (Ctr). In total, we have scanned 26 samples, all whichwihch had undergone routine histopathological assessment beforehand. We used both a synchrotron holo-tomography setup and a laboratory µCT with custom designed instrumentation and reconstruction workflow, as described in *Eckermann et al., 2020*. Based on the reconstructed volume data, we then determined structural parameters, such as the orientation of the cardiomyocytes and the degree of anisotropy, as well as a set of shape

measures defined from the structure tensor analysis. This procedure is already well established for Murine heart models (*Dejea et al., 2019*). Segmentation of the vascular network enabled by deep-learning methods is used to quantify the architecture of the vasculature.

Following this introduction, we describe the methodology, which is already summarized in *Figure 1*. We then describe the reconstructed tissue data in terms of histopathological findings based on visual impression, and compare the different groups. We then apply automated image processing for classification and quantification of tissue pathologies. Finally, we segment the vasculature using a deep-learning-based approach based on sparse annotations and quantify the structure of the capillary network by graph representations of the segmented vessels and quantify the vasculature, both from voxel-based measures and from extracted graph representations of the segmented vessel network. From the generalized shape measures, as well as the inspection of particular vessel architectures exhibiting the IA phenomenon, distinct changes of Cov with respect to the other pathologies and to Ctr are observed. The paper closes with a short conclusions and outlook section.

## Materials and methods
### Autopsy, clinical background, and tissue preparation
In total, 26 postmortem heart tissue samples were investigated: 11 from Covid-19 patients (Cov), 4 from H1N1/A influenza patients (Inf), 5 from patients who died due to coxsackie virus myocarditis (Myo), as well as 6 control samples (Ctr). The age and sex of all patients are summarized in *Table 1*. Detailed information about age, sex, cause of death, hospitalization, clinical, radiological, and histological characteristics of all patients is given in Appendix 2 Tab. D.1.

*Figure 1* illustrates the sample preparation and the tomographic scan geometries used to assess the 3d cytoarchitecture on different length scales. FFPE-tissue from autopsies was prepared by standard formalin fixation and paraffin embedding. From the paraffin-embedded tissue block, sections of $3\,\mu$m thickness were prepared for histomorphological assessment using conventional hematoxylin and eosin (HE) staining. One representative microscopy image of a Covid-19 patient is shown in *Figure 1*. An overview of HE stained sections from all samples is shown in the *Appendix 1—figure 1*. In previous studies, we could show the correlation of 3d X-ray phase contrast tomography data with conventional 2d histology (*Eckermann et al., 2020*; *Töpperwien et al., 2018*).

Biopsy punches with a diameter of $3.5\,$mm were then taken and transferred onto a holder for the tomographic scans. All samples were first scanned at a laboratory μCT instrument using a liquid metal jet anode. Next, tomograms of Covid-19 and control samples were scanned at the GINIX endstation of the P10 beamline at the PETRAIII storage ring (DESY, Hamburg), using the parallel (unfocused) synchrotron beam. Finally, biopsy punches with a diameter of $1\,$mm was taken from the $3.5\,$mm biopsy of one control and one Covid-19 sample and scanned at high geometric magnification $M$ using a cone beam illumination emanating from a X-ray waveguide (WG).

### Tomographic recordings
#### Liquid metal jet (LJ) setup
All samples were scanned using a home-built laboratory phase-contrast μCT-setup, as sketched in *Figure 1C*. X-rays emitted from a liquid metal jet anode (Excillum, Sweden) are used in cone beam geometry with a geometric magnification $M = \frac{x_{01}+x_{12}}{x_{01}}$ controlled by the source-sample $x_{01}$ and sample-detector distance $x_{12}$. The spectrum of photon energy $E$ is dominated by the characteristic Kα lines of galinstan ($Ga, Zn, In$ alloy), in particular the $Ga$ line $E_{Ga} =9.25\,$keV. Projections were acquired by a sCMOS detector with a pixel size of $px = 6.5\,\mu$m coupled by a fiber-optic to a 15-m-thick Gadox-scintillator (Photonic Science, UK) (*Bartels et al., 2013*; *Reichardt et al., 2020*). In this work, data was acquired at an effective pixel size of $px_{\mathrm{eff}} = \frac{px}{M} = 2\,\mu$m. For each of the 1,501 angular positions 3 projections at 0.6 s acquisition time were averaged. Further, 50 flat field images before and after the tomographic acquisition, as well as 50 dark field images after the scan were recorded. The total scan time was approximately one hour per sample.

#### Parallel beam (PB) setup
All Cov and Ctr samples were also scanned with an unfocused, quasi-parallel synchrotron beam at the GINIX endstation, at a photon energy $E_{\mathrm{ph}}$ of $13.8\,$keV. Projections were recorded by a microscope

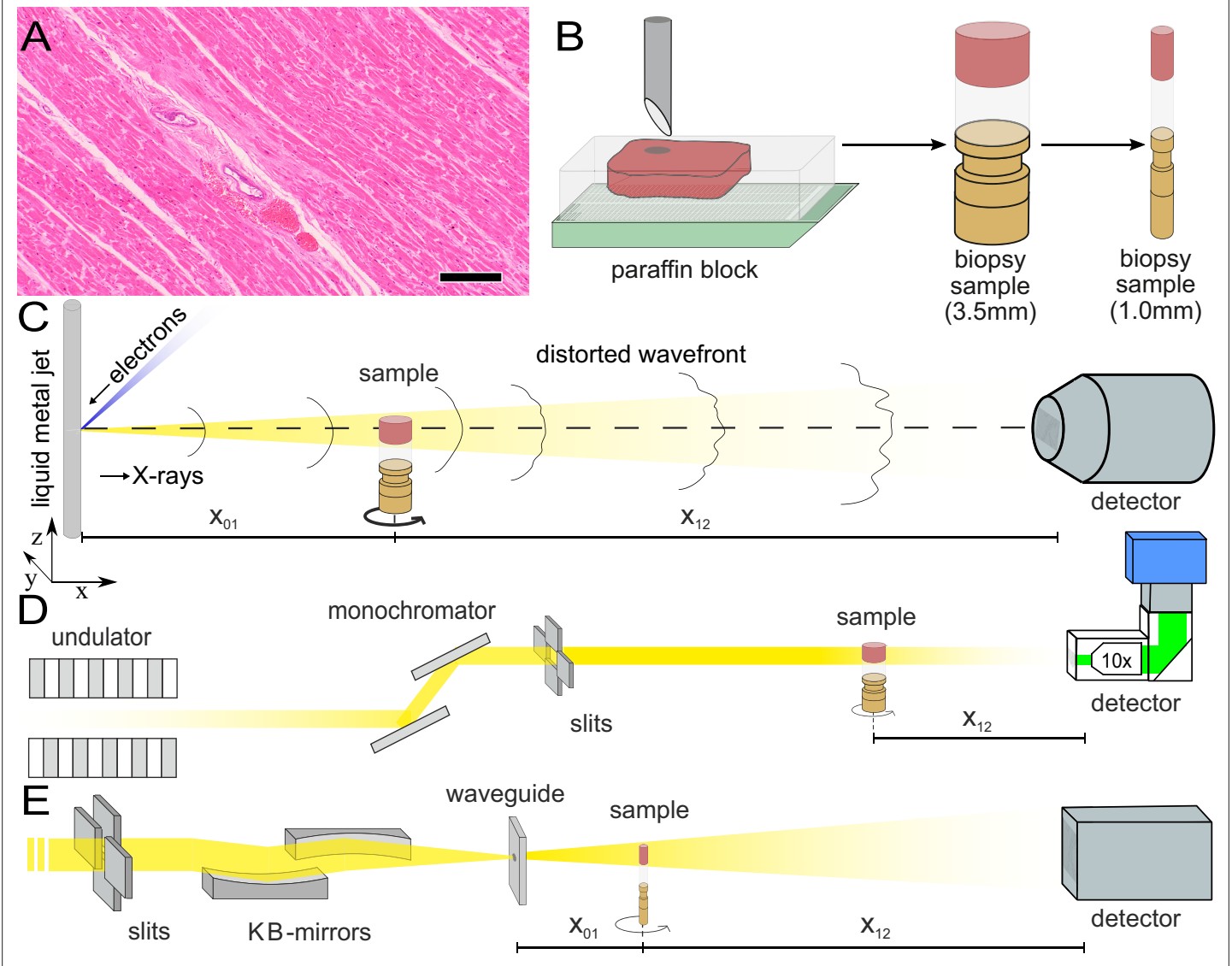

**Figure 1.** Sample preparation and tomography setups. (**A**) HE stain of a 3-m-thick paraffin section of one sample from a patient who died from Covid-19 (Cov-I, Scalebar: $100\,\mu$m). In total, 26 postmortem heart tissue samples were investigated: 11 from Covid-19 patients, 4 from influenza patients, 5 from patients who died with myocarditis and six control samples. (**B**) From each of the samples, a biopsy punch with a diameter of $3.5\,$mm was taken and transferred onto a holder for the tomography acquisition. After tomographic scans of all samples at the laboratory setup, Covid-19 and control specimens were investigated at the synchrotron. Furthermore,at the laboratory and parallel beam setup at the synchrotron, one punch with a diameter of $1\,$mm was taken from one of the control and Covid-19 samples for investigations at high resolution. (**C**) Sketch of the laboratory micro-CT setup. Tomographic scans of all samples were recorded in cone beam geometry with an effective pixel size of $px_{\mathrm{eff}} = 2\,\mu$m using a liquid metal jet source (EXCILLUM, Sweden). (**D**) Sketch of the parallel beam setup of the GINIX endstation (P10 beamline, DESY, Hamburg). In this geometry, datasets of Covid-19 and control samples were acquired at an effective voxel size of $650\,$nm$^3$. One plane of each sample was covered by 3×3 tomographic recordings. For each sample a plane of 3×3 tomographic acquisitions was recorded. (**E**) Cone beam setup of the GINIX endstation. After the investigation in parallel geometry, the 1 mm biopsy punches of one control and Covid-19 sample were probed and a high resolution scan in cone beam geometry was recorded. This configuration is based on a coherent illumination by a wave guide and allows for high geometric magnification and effective voxel sizes below $200\,$nm.

detection system (Optique Peter, France) with a 50-m-thick LuAG:Ce scintillator and a 10× magnifying microscope objective onto a sCMOS sensor (pco.edge 5.5, PCO, Germany) (**Frohn et al., 2020**). This configuration enables a field-of-view (FOV) of $1.6\,$mm, sampled at a pixel size of $650\,$nm. The continuous scan mode of the setup allows to acquire a tomographic recording with 3000 projections over 360° in less than 2 min.3×3 tomographic acquisitions in one plane for each of the 3.5 mm biopsy punches. For each sample, one plane of the 3.5 mm biopsy punch was covered by 3×3 tomographic acquisitions.

**Table 1.** Sample and medical information of patients.

| Sample group | N patients | Sample quantity | Age | Sex |
|---|---|---|---|---|
| Control | 2 | 6 | 31 ± 7 | 2 F |
| Covid-19 | 11 | 11 | 76 ± 13 | 10 M, 1 F |
| Myocarditis | 5 | 5 | 43 ± 17 | 4 M, 1 F |
| Influenza | 4 | 4 | 63 ± 9 | 3 M, 1 F |

Afterwards, dark field and flat field images were acquired. In total more than 150 tomographic scans (nine tomograms for each of the 17 samples) were recorded in this configuration.

## Waveguide (WG) setup

As a proof-of-concept that subcellular resolution can also be obtained on cardiac tissue samples, a 1 mm-diameter biopsy punch was taken from both a Covid-19 and control sample, both of which were previously scanned (PB geometry). The highly coherent cone beam geometry and clean wavefront of the WG illumination allows for samples to be probed at high magnification in the holographic regime. Here, the sample was aligned at $M \simeq 40$, resulting in an effective pixel size of 159 nm. Images of the Ctr were acquired by a sCMOS Camera (15 $\mu$m Gadox scintillator, 2560 ×2,160 pixel) with a physical pixel size of 6.5 m (Andor Technology Ltd, UK). Cov datasets were recorded by a 1:1 fiber-coupled scintillator-based sCMOS camera (2048 × 2048 pixels, Photonic Science, Sussex, UK) with a custom 15-m-thick Gadox scintillator with pixel size of 6.5 $\mu$m. For Ctr data, the photon energy was $E =$ and 1500 projections over 180 degrees were recorded with an acquisition time of 0.3 s, for the Cov sample 1500 projections were acquired for four slightly different propagation distances at $E =$. The difference in acquisition time of both scans (Ctr: $\simeq$ 10 min, Cov $\simeq$ 3 h) is given by different wavguide channel diameters and guiding layer materials (Ctr: Ge, Cov: Si). Before and after each tomographic scan, 50 empty beam projections as well as 20 dark fields after the scan were recorded. The experimental and acquisition parameters for all imaging modalities are listed in **Table 2**.

### Phase retrieval and tomographic reconstruction

The 3d structure of the cardiac tissue was reconstructed from the raw detector images. To this end, we computed the phase information of each individual projection and performed tomographic reconstruction to access the 3d electron density distribution. For image processing and phase retrieval, we used the HoloTomoToolbox developed by our group, and made publicly available (**Lohse et al., 2020**) (**Table 3**). First, flat field empty beam and dark field corrections were performed for all raw projections. In addition, hot pixel and detector sensitivity variations were removed by local median filtering. Phase

**Table 2.** Data acquisition parameters of the laboratory and synchrotron scans.

| Parameter | LJ setup | PB setup | WG setup (Ctr/Cov) |
|---|---|---|---|
| Photon energy (keV) | 9.25 | 13.8 | 10/10.8 |
| Source-sample-dist. $x_{01}$ (m) | 0.092 | $\simeq$ 90 | 0.125/0.125 0.127 0.131 0.139 |
| Sample-detector-dist. $x_{12}$ (m) | 0.206 | 0.5 | 4.975 |
| Geometric magnification M | $\simeq$ 3 | $\simeq$ 1 | $\simeq$ 40 |
| Pixel size (µm) | 6.5 | 0.65 | 6.5 |
| Effective pixel size (µm) | 2 | 0.65 | 0.159 |
| Field-of-view h× v (mm²) | 4.8×3.4 | 1.6× 1.4 | 0.344× 0.407/0.325× 0.325 |
| Acquisition time (s) | 3× 0.6 | 0.035 | 0.3/2.5 |
| Number of projections | 1501 | 3000 | 1500 |
| Number of flat field empties | 50 | 1000 | 50 |
| Number of dark field | 50 | 150 | 20 |

**Table 3.** Phase retrieval algorithms and parameters used for the different setups.

| Setup | LJ setup | PB setup configuration | WG setup configuration |
|---|---|---|---|
| Fresnel number | 0.47125 | 0.0095 | 0.0017 |
| phase retrieval | BAC | CTF | nonlinear CTF |
| $\delta/\beta$-ratio | - | 1/45 | 1/130 |
| parameter | $\alpha = 8 \cdot 10^{-3}$ | $\alpha_1 = 10^{-3}$ | $\alpha_1 = 8 \cdot 10^{-4}$ |
| | $\gamma = 1$ | $\alpha_2 = 0.5$ | $\alpha_2 = 0.2$ |

retrieval of LJ scans was carried out with the Bronnikov aided Correction (BAC) algorithm (*De Witte et al., 2009*; *Töpperwien et al., 2018*). For the PB scans, a local ring removal (width of ±40 pixel) was applied around areas where wavefront distortions from upstream window materials did not perfectly cancel out after empty beam division. Phase retrieval of PB scans was performed using the linear CTF-approach (*Cloetens et al., 1999*; *Turner et al., 2004*). Phase retrieval of WG scans was performed using a nonlinear approach of the CTF. This advanced approach does not rely on the assumption of a weakly varying phase, and iteratively minimizes the Tikhonov-functional starting from the CTF result as an initial guess. For a weakly phase-shifting sample (linear approximation) without further constraints, both approaches yield exactly the same result (*Lohse et al., 2020*). Apart from phase retrieval, the HoloTomoToolbox provides auxiliary functions, which help to refine the Fresnel number or to identify the tilt and shift of the axis of rotation (*Lohse et al., 2020*). Tomographic reconstruction of the datasets was performed by the ASTRA toolbox (*van Aarle et al., 2016*; *van Aarle et al., 2015*). For the LJ and WG scans recorded at large cone beam geometry, the FDK-function was used, while the PB was reconstructed by the iradon-function with a Ram-Lak filter.

To combine the 3×3 tomographic volumes, covering one plane of the 3.5 mm biopsy in PB geometry, a non-rigid stitching tool of was used (*Miettinen et al., 2019*). Region-of-interest artefacts of the PB reconstructions, which led to circular low frequency artefacts at the borders of the biopsy reconstruction volume, were removed by radial fitting of cosine functions. In order to decrease the size of the stitched volume, and thus also reduce computational power needed for further analysis, the datasets were binned by a factor of 2.

## Structure tensor analysis

The laboratory datasets and the stitched datasets reconstructed from the PB recordings were used for further analysis of the cardiac structure, cytoarchitecture and the corresponding pathological changes, see the workflow sketched in *Figure 2*. For each reconstruction of the 3d electron density map (*Figure 2A*), the biopsy punches were first masked based on their higher electron density compared to the surrounding air. Missing areas in the PB acquisition (from corrupted datasets) were excluded. The intensities of the reconstructions were normalized. *Figure 2B* shows an exemplary masked 2D slice. For each sample, the local tissue orientation and the degree of alignment was then determined from structure tensor analysis (*Krause et al., 2010*). Accordingly, the local structural orientation at point $\mathbf{r}$ can be described by a vector $w$

$$w(\mathbf{r}) = \mathrm{argmin}_{v=1}(I(\mathbf{r}+v) - I(\mathbf{r}))^2 \tag{1}$$

with $v \in \mathbb{R}^3$ and $|v| = 1$ in voxel units. Since the vector $w$ or set of vectors is computed from partial derivatives, one has to first compensate for the ill-posedness of computing derivatives of noisy intensity values by convolving intensities $I_\sigma = \mathcal{K}_\sigma * I$ with a Gaussian kernel $\mathcal{K}_\sigma$. The structure tensor $J_\rho$ then is defined as follows

$$J_\rho = \mathcal{K}_\rho * \begin{pmatrix} (\partial_x I_\sigma)^2 & (\partial_x I_\sigma)(\partial_y I_\sigma) & (\partial_x I_\sigma)(\partial_z I_\sigma) \\ (\partial_y I_\sigma)(\partial_x I_\sigma) & (\partial_y I_\sigma)^2 & (\partial_y I_\sigma)(\partial_z I_\sigma) \\ (\partial_z I_\sigma)(\partial_x I_\sigma) & (\partial_z I_\sigma)(\partial_y I_\sigma) & (\partial_z I_\sigma)^2 \end{pmatrix}, \tag{2}$$

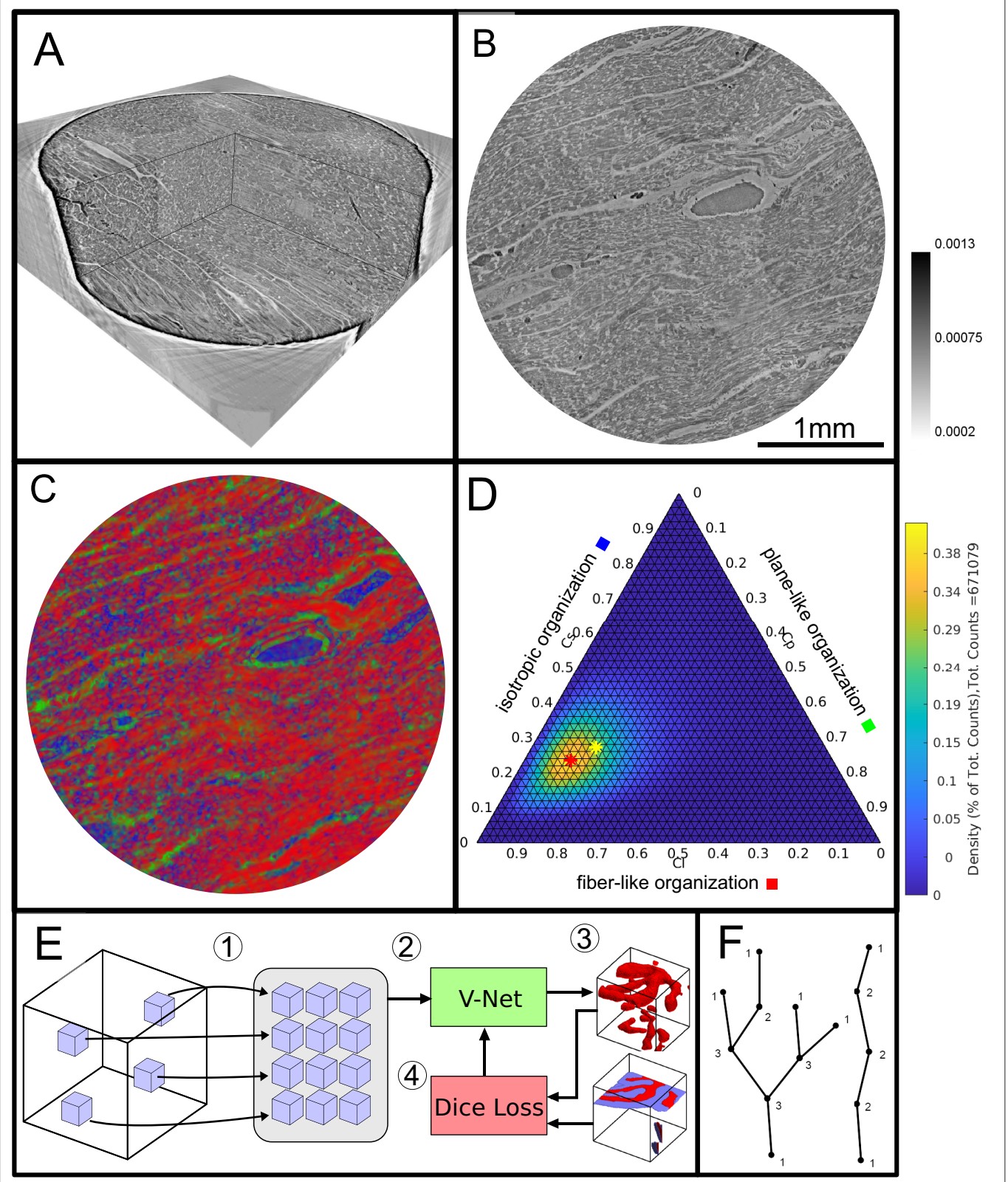

**Figure 2.** Data analysis workflow of cardiac samples. (**A**) Volume rendering of a tomographic reconstruction from PB data. (**B**) Orthogonal slice of the masked tissue. Scale bar: 1 mm (**C**) Shape measure distribution ($C_l$ red, $C_p$ green and $C_s$ blue) of the slice shown in B. (**D**) Ternary plot of shape measure distribution. The peak (red) and mean (yellow) values are marked with an asterisk. (**E**) Overview of the training process for the neural network. (1) Random subvolumes (containing labeled voxels) are sampled from the full volume and are collected in a batch. (2) The batch is fed through the neural network,

*Figure 2 continued on next page*

*Figure 2 continued*

resulting in (3) a segmentation (top) and labels for one subvolume (bottom). (4) The dice loss is computed from segmented subvolumes based on labeled voxels, and the parameters of the neural network are updated. (**F**) Scheme of branching and the relation to degree of the vessel nodes obtained by a graph representation of the segmented microvasculature.

---

where a second convolution $\mathcal{K}_\rho$ is applied with length scale $\rho$, thus defining the structural scale on which the tissue structure is analyzed/reported. Since the reconstructed electron density $I(\mathbf{r})$ along a fiber is approximately constant along the fiber tangent, the vector describing the local structural orientation is given by the eigenvector with the smallest eigenvalue of the symmetric matrix $J_\rho$. The implementation of the structure tensor analysis is provided in https://lab.compute.dtu.dk/patmjen/structure-tensor. In this work, the size of $\rho$, determining the sub-volume on which the structural analysis is performed, was set to 32 pixels for PB datasets and 12 pixels for LJ acquisitions. This corresponds to $\approx 20.8\mu$m and $24\mu$m, respectively, that is a value slightly smaller than the diameter of a cardiomyocyte ($\approx 25\mu$m). A smoothing parameter $\sigma$ of 2 pixels was chosen to reduce noise. From the eigenvalues ($\lambda_1 \geq \lambda_2 \geq \lambda_3$) of $J_\rho$, quantitative shape measures (as first introduced for diffusion tensor MRI data) can be determined (*Westin et al., 2002*). These parameters describe the degree of anisotropy of the local structure orientation. Tissue structure with fiber-like symmetry are indicated by a high value of

$$C_l = \frac{\lambda_2 - \lambda_3}{\lambda_1} \ . \tag{3}$$

Plane-like (lamellar) symmetry is described by a high value of

$$C_p = \frac{\lambda_1 - \lambda_2}{\lambda_1} \ , \tag{4}$$

and isotropic structures are described by a high value of the spherical shape measure

$$C_s = \frac{\lambda_3}{\lambda_1} \ . \tag{5}$$

The shape measure distribution of the exemplary slice is shown in *Figure 2C*. Red areas indicate a high $C_l$ value and correlate with the well aligned chains of cardiomyocytes. Planar structures as collagen sheets and separated muscle bundles show a high $C_p$ value and are color-coded in green. Isotropic areas as blood filled vessels are represented by a high $C_s$ value (blue). The values of the three measures range between zero and one, and sum up to one

$$C_l + C_p + C_s = 1 \ . \tag{6}$$

Thus, one of the three shape measures is redundant. The data can be plotted in a ternary diagram as used to represent phase diagrams of ternary mixtures (see *Figure 2D*). To characterize the distribution of the shape measures for each sample, a principal component analysis (PCA) was performed. Note, that for the LJ datasets, the paraffin surrounding the cardiac tissue was removed by an intensity-based mask. Since one axis of the shape measure is redundant, the distribution of all data points can be described by two eigenvectors ($\mathbf{u_1}, \mathbf{u_2}$ with the largest eigenvalues ($\eta_1, \eta_2$)). The PCA analysis is equivalent to a two-dimensional Gaussian with standard deviation $\sqrt{\eta_1}, \sqrt{\eta_2}$. The two eigenvectors ($\mathbf{u_1}, \mathbf{u_2}$) can be represented by the major and minor axis of an ellipse centred around the mean ($\mu_l, \mu_p, \mu_s$) (yellow asteroid) representing the 'point cloud' of all shape measures. The eccentricity of the ellipse is given by

$$e = \sqrt{1 - \frac{\sqrt{\eta_2}}{\sqrt{\eta_1}}} \tag{7}$$

and describes how much the ellipse deviates from being circular. The area of the ellipse is given by $A_\eta = \pi\sqrt{\eta_1\eta_2}$ and is a measure for the dispersion of the shape measure distribution. The eccentricity indicates whether the dispersion is isotropic in the plane of the shape parameters. Large values of $e$ indicate a sharp elongated distribution along the major axis of the ellipse.

## Segmentation by deep learning

A deep learning approach based on the V-Net architecture (*Milletari et al., 2016*) was used to segment the vascular network in the PB datasets. The V-Net can be regarded as a 3D version of the popular U-Net architecture (*Ronneberger et al., 2015*) often used for segmentation of medical images. Training was performed using the Dice loss (*Milletari et al., 2016*) and the ADAM optimizer (*Kingma and Ba, 2015*) with step size 0.001 and hyperparameters $\beta_1 = 0.9$ and $\beta_2 = 0.999$. To avoid the need of a fully labeled training dataset, a training strategy using sparsely annotated data sets was adopted, inspired by *Çiçek et al., 2016*. In each dataset, a small number of axis-aligned 2D slices was annotated manually, and the Dice loss was evaluated only for these annotated voxels. Prior to training, the annotated volumes were split into a training set and a smaller validation set. The network was trained on the training set, while the quality of the current model (network weights) was tested on the validation set, as sketched in *Figure 2E*. Instead of segmenting the entire volume before computing the loss, batches of 12 random subvolumes of size $96 \times 96 \times 96$ voxels were selected, ensuring that each contained annotated voxels. These were then fed into the network, the loss was computed, and the parameters (network weights) were updated. After running on 256 subvolumes, the network was evaluated by running it on the validation set. Rotations by 90 degrees and mirror reflections (axis flips) were used both on the training and the validation subvolumes to augment the data. The neural network code of this implementation was uploaded to GitHub (github.com/patmjen/blood-vessel-segmentation; *Jensen, 2021* copy archived at swh:1:rev:783df24c3068e35f2ae994cab095b4318c755b29).

A separate model was trained for a Covid-19 volume (Cov-IV) and a control volume (Ctr-III). The models were trained for 24 hr (~900 epochs) using an NVIDIA Tesla V100 32 GB GPU, and the model version which achieved the highest validation score during the training was kept. Finally, the training was performed over two rounds. First, an initial training and validation set was created to train the model. Then, the training set was improved by adding additional annotations to areas which were falsely segmented, and a new model was trained on the improved data.

As the segmentation masks produced by the neural networks typically contained a number of errors, a post-processing pipeline was designed to reduce the errors' effect. The first step is to reduce the number of false positives. These typically materialize as small, roughly spherical regions of background which was erroneously detected as blood vessels. To remove them, the structure tensor shape measures $C_l$, $C_p$, and $C_s$ are computed for the segmentation mask (treating background as 0 and foreground as 1) with $\sigma$ and $\rho$ set to 1 and 8 voxels, respectively. Then, all connected components with a volume less than $10^4$ voxels or a mean value of $C_s$ greater than 0.2 are removed. The thresholding on $C_s$ ensures that isotropic components are removed regardless of their size while still preserving smaller sections of correctly segmented blood vessels. The last step is to reduce the number of false negatives by reconnecting segments of blood vessels which are disconnected due to small errors in the segmentation. Since endpoints of blood vessels will typically have a large value of $C_s$, small gaps in the vessels can be closed by performing a morphological closing of the isotropic regions of the segmentation mask. Specifically, the cleaned binary mask, $\hat{B}$, is given by

$$\hat{B} = \max(B, \text{close}(C_l \odot B, S_4) > 0.2) , \tag{8}$$

where $B$ is the original binary mask (after the first post processing step), $C_l$ is the line-like measure for all voxels in $B$, and $\text{close}(C_l \odot B, S_4)$ denotes a closing of the elementwise product between $C_l$ and $B$ with a ball of radius 4. For performance reasons the closing uses an approximated ball as described in *Jensen et al., 2019*.

## Quantification of the vascular system

A quantitative description of the vascular system was achieved by modeling the segmented vessels as a mathematical graph. A graph consists of a set of vertices and a set of edges where each edge connects a pair of vertices. If vertices are connected via an edge they are said to be neighbors and the degree of a vertex (nodes) $n$ is equal to its number of neighbors. In *Figure 2F* a sketch of a vessel graph is shown for a straight vessel and for a vessel with multiple branching points. The degree of connectivity $n$ is added to the sketch. This gives a natural correspondence to the complex vascular system by modeling bifurcation points as vertices and the blood vessels between pairs of bifurcation points as edges. Furthermore, structural phenomena such as excessive vessel bifurcation and intussusceptive angiogenesis can now be detected by, respectively, a large number of high degree

vertices and loops in the graph. The graph corresponding to the vascular system is extracted from the segmentation created by the neural network. First, a skeletonization (*Lee et al., 1994*) is computed, which reduces all structures in the binary volume to 1-voxel wide centerlines without changing the connectivity. These centerlines are then converted to a graph as described in *Kollmannsberger et al., 2017*. Once the graph is constructed the vertex degrees can readily be extracted by counting the number of edges connected to each vertex. Loops are detected using the algorithm from *Gashler and Martinez, 2012* which detects all atomic cycles in a given graph. A cycle is a path through the graph that begins and ends at the same vertex without reusing edges. An atomic cycle is a cycle which cannot be decomposed into shorter cycles. Only reporting atomic cycles is more robust, since small errors in the segmentation may cause the skeletonization to contain long cycles that do not correspond to anatomical structures. The 3d data sets (including tomographic reconstructions and segmentations) was visualized using the software Avizo (Thermo Fisher Scientific).

## Vascular corrosion casting, scanning electron microscopy, and morphometry

The microvascular architecture of Covid-19 hearts was also examined using scanning electron microscopy (SEM) and microvascular corrosion casting. So far, corrosion casting coupled with SEM represents the gold standard for assessing the subtypes of angiogenesis. The afferent vessels of heart specimens were cannulated with an olive-tipped cannula. The vasculature was flushed with saline (at body temperature) followed by glutaraldehyde fixation solution (2.5%, pH 7.4, Sigma Aldrich, Munich, Germany). Fixation was followed by injection of prepolymerized PU4ii resin (VasQtec, Zurich, Switzerland) mixed with a hardener (40% solvent) and blue dye as casting medium. After curing of the resin, the heart tissue was macerated in 10% KOH (Fluka, Neu-Ulm, Germany) at 40°C for 2–3 days. Specimens were then rinsed with water and frozen in distilled water. The casts were freeze-dried and sputtered with gold in an argon atmosphere and examined using a Philips ESEM XL-30 scanning electron microscope (Philips, Eindhoven, Netherlands). Vascular morphometry of variants of angiogenesis were then assessed: high-power images of the capillary network were scanned and quantified.

# Results and discussion
## Reconstructed electron density: laboratory data

*Figure 3* shows representative slices of the tomographic reconstruction for all samples scanned at the laboratory LJ setup. The image quality is sufficient to identify the cytoarchitecture and main structural features of interest, such as the general orientation of the cardiomyocytes, large arteries and veins, as well as smaller capillaries. Occasionally, artefacts from sample preparation, such as small air filled micro-fractures of the paraffin, also appear in the reconstructions. In the top row of *Figure 3*, two annotated slices representative for the Covid-19 and control group are shown enlarged. In the following, the structural appearance of the different groups (Ctr, Cov, Inf and Myo) is briefly described.

### Control (Ctr)

The reconstructions of the control hearts are shown in the top row (*Figure 3* (Ctr-I to Ctr-VI)). Biopsies Ctr-I to Ctr-III and Ctr-IV to Ctr-VI were taken from different areas of the same heart, respectively. In general, the cardiac structure with interload cardiomyocytes and vasculature of the control group is well preserved. The cardiomyocytes are arranged in close proximity and form bundled elongated myocyte chains. Vessels appear as bright tubes within the dense, homogeneous muscle tissue and only a few blood residues can be found in the vessels. Ctr-III differs from the other control samples. The alignment of the cardiomyocytes is not directed along the same direction, and the amount of collagen sheets and paraffin inclusions is comparably high. Further, a high amount of adipose tissue can be identified, as accumulations of less electron-dense (i.e. brighter) spheroids, see for example the top of the slice. Ctr-III also shows a high amount of collagen sheets, which appear as dark stripes in the reconstructions. Ctr-V contains many electron-dense spheres.

### Covid-19 (Cov)

The cardiac samples of the hearts from patients who died from Covid-19 are shown in the next two rows of *Figure 3* (Cov-I to Cov-XI). Compared to Ctr, all Cov samples show a high amount of blood filled, ectatic vessels with abrupt changes in diameter, plausibly correlating to micro-thrombi. The

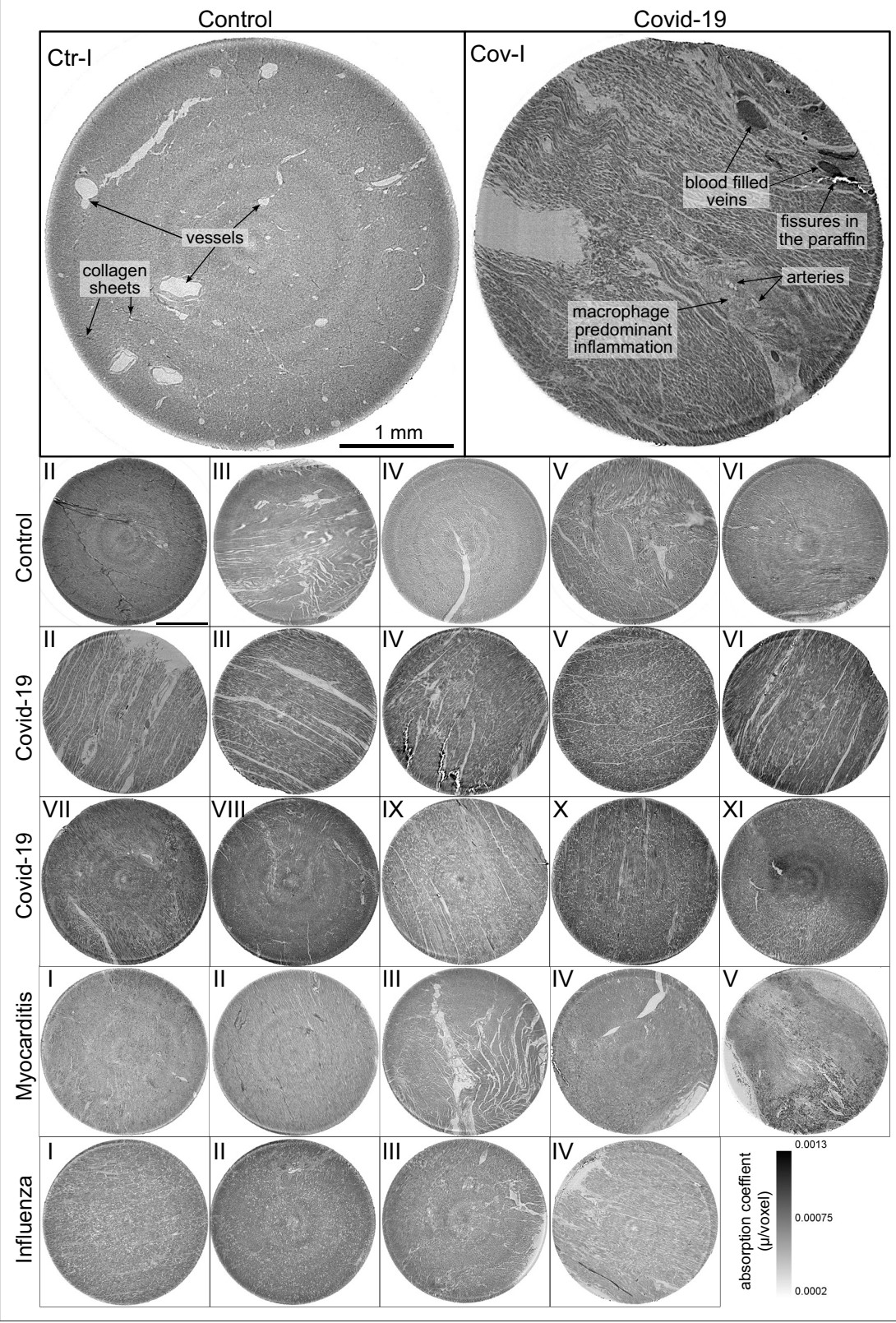

**Figure 3.** Overview of reconstruction volumes: Laboratory setup. For each sample analyzed at the LJ μ-CT setup one slice of the reconstructed volume is shown. In the top row, a slice of a tomographic reconstruction of a control sample (Ctr-I) and of a sample from a patient who died from Covid-19 (Cov-I) are shown. Below, further slices from control (Ctr-II to Ctr-VI), Covid-19 (Cov-II to Cov-XI) as well as myocarditis (Myo-I to Myo-V) and influenza (Inf-I to Inf-IV) samples are shown. Scale bars: 1 mm.

cardiomyocytes are not densely packed with substantial interstitial edema, and correspondingly there is a high amount of paraffin inclusions between the cells. This may also explain a higher amount of micro-fractures (e.g. in Cov-I and IV) in the paraffin, which are filled with air. Furthermore, Cov-I also shows an inflammatory infiltrate, predominantly consisting of macrophages, around the intramyocardial vessel, marked in the corresponding slice (top, right) of *Figure 3*.

### Coxsackie virus myocarditis (Myo)
In *Figure 3*, representative slices from tomographic reconstructions of biopsies of patients who died from coxsackie myocarditis (Myo-I to Myo-V) are shown. The tissue of the Myo group is almost as densely packed as the Ctr group. Only in the biopsy of Myo-III, which was sampled near an artery, some large paraffin inclusions between the cardiomyocytes are visible. Characteristic for all myocardits samples is a high amount of lymphocytes, which appear as small electron-dense spheres in the reconstructions. They are primarily located close to vessels (as in Myo-II), but also appear inside the ECM between cardiomyocytes (Myo-I), or infiltrate extensive areas of tissue devoid of vital cardiomyocytes, corresponding to necrosis (Myo-V).

### Influenza (Inf)
The biopsies taken from patients who succumbed to H1N1/A influenza (Inf-I to Inf-IV) are shown in the bottom row of *Figure 3*. The tissue structure in this group is also densely packed. Inf-IV shows a high amount of blood filled vessels with abrupt changes in caliber, plausibly correlating to micro-thrombi. Otherwise, changes include lymphocytic infiltration and regions devoid of vital cardiomyocytes indicating necrosis, similar to (Myo).

In summary, the quality of the reconstructions from laboratory data was already sufficiently high to identify the main anatomical features of the cardiac tissue, readily by eye in selected slices. The full reconstruction volumes were therefore targeted by automated geometric analysis based on a structure tensor approach, as described in the next section. However, smaller capillaries and subcellular features were not resolved in the laboratory LJ setup configuration. Thus, imaging using high coherent synchrotron radiation was chosen to analyze vascular changes within the tissue.

## Reconstructed electron density: synchrotron data
### PB setup configuration
The samples from Ctr and Cov patients were scanned in the PB setup configuration of the GINIX endstation (Hamburg, DESY). Compared to the laboratory acquisitions, this allowed for smaller effective voxel sizes and enabled a higher contrast for smaller tissue structures as erythrocytes and capillaries (as shown in AppendixC Fig. C). Slices of the tomographic reconstruction of the 3d electron density distribution are shown in AppendixC Fig. C and were used for the segmentation of the vascular system.

### WG setup
In order to further explore high-resolution imaging capabilities, tomograms of two selected biopsies (Ctr-VI and Cov-III) with a diameter of 1 mm were recorded at the WG setup of the GINIX endstation, exploiting cone beam magnification and high coherence filtering based on the waveguide modes. *Figure 4* shows the corresponding results. A cut of the entire control volume with a size of about 340 × 340 × 400 m$^3$ is shown in *Figure 4*. *Figure 4* shows a slice through the tomographic reconstruction perpendicular to the orientation of the cardiomyocytes. A closer inspection of a single cardiomyocyte marked with a red box is shown on the right. The nucleus of the cell with nucleoli can be clearly seen. Within the cytosol, the myofibrils appear as small discs in the slice. *Figure 4C* shows a second slice through the 3d volume which is oriented along the orientation of the cardiomyocytes. In this view, intercalated discs can be identified. They appear as dark lines connecting two cardiomyocytes. A magnification of the area is marked with a red box. In this view, the myofibrils can be identified as elongated lines within the cell. This region also contains a nucleus of one cardiomyocyte, but also an intercalated disc at the bottom of the image. The tomographic reconstruction of the Cov sample is shown in the lower part of *Figure 4* in the same manner as the Ctr. In this dataset capillaries, nuclei and myofibrils can also be identified. The volume contains smaller capillaries compared to the control, but this circumstance is probably due to a different location within the myocardium. The most

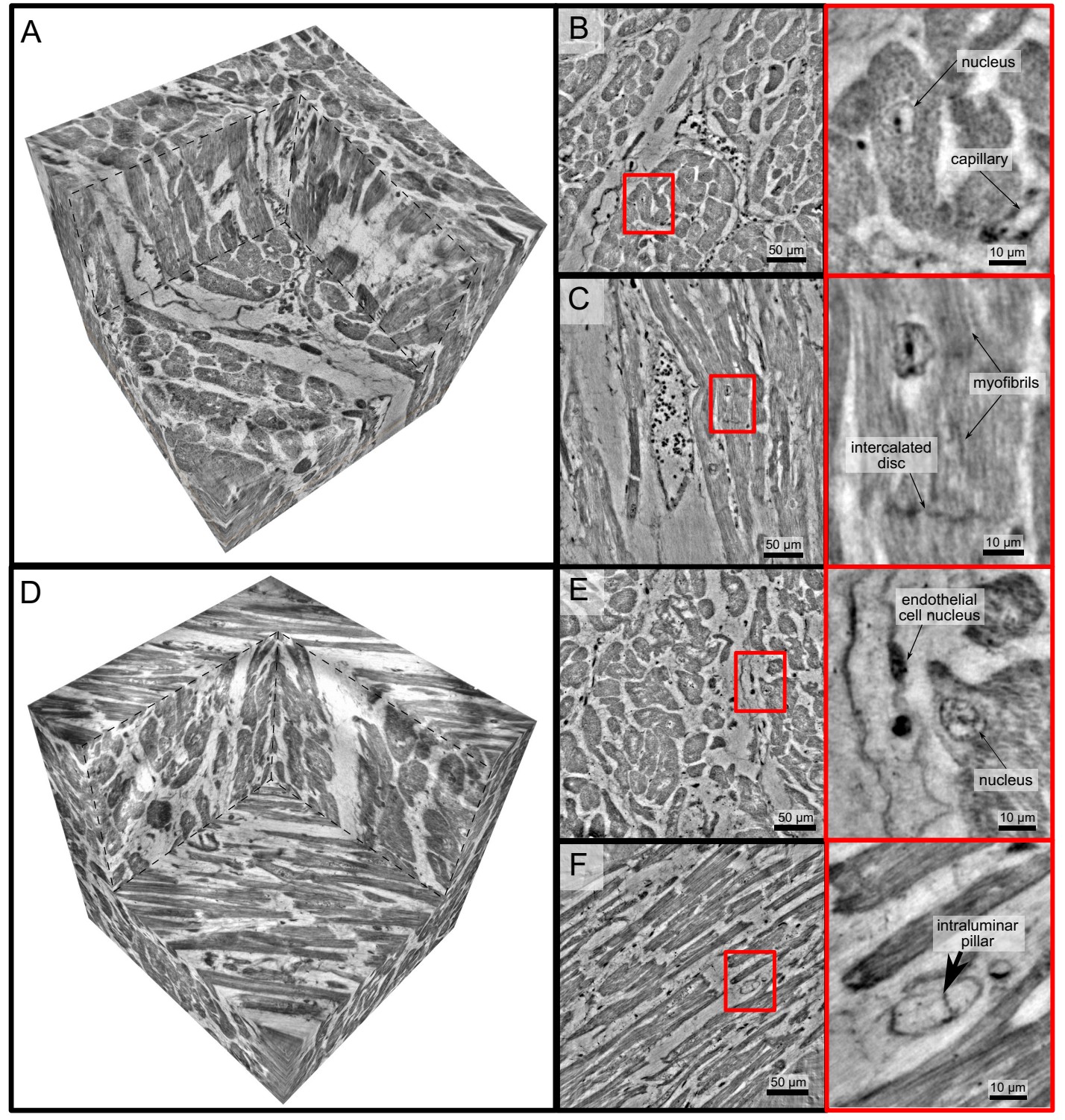

**Figure 4.** High-resolution tomogram of cardiac tissue recorded in cone beam geometry. (**A**) Volume rendering of a tomographic reconstruction from a control sample recorded in cone beam geometry based on a wave guide illumination. After the analysis in parallel beam geometry, a biopsy with a diameter of **1 mm** was taken from the **3.5 mm** biopsy punch. This configuration revealed sub-cellular structures such as nuclei of one cardiomyocytes, myofibrils and intercalated discs. (**B**) Slice of the reconstructed volume perpendicular to the orientation of the cardiomyocytes. The red box marks an area which is magnified and shown on the right. One cardiomyocyte is located in the center of the magnified area. In this view, the nucleus can be identified. It contains two nucleoli, which can be identified as dark spots. The myofibrils appear as round discs. (**C**) Orthogonal slice which oriented along the orientation of the cardiomyocytes. A magnification of the area marked with a red box. In this view, a nucleus but also the myofibrils can

*Figure 4 continued on next page*

*Figure 4 continued*
be identified as dark, elongated structures in the cell. Further, an intercalated disc is located at the bottom of the area. (**D**) Volume rendering of a tomographic reconstruction from a Covid-19 sample. Slices orthogonal (**E**) and along (**F**) to the cardiomyocyte orientation are shown on the right. In the magnified areas, a nucleus of an endothelial cell and an intraluminar pillar -the morphological hallmark of intussusceptive angiogenesis- are visible. Scale bars: orthoslices $50\,\mu$m; magnified areas $10\,\mu$m.

important difference between the Ctr and Cov sample is the presence of small bars in the lumen of capillaries in the Cov sample. These intraluminal pillars are an indicator for IA.

Since the FOV in this configuration is limited, and stitching of larger volumes required more beamtime than available, quantitative and statistical analysis was performed only on the datasets acquired in the laboratory and in PB geometry. At the same time, this proof-of-concept shows that much more structural information could be exploited by stitching tomography and speeding-up the measurement sequence in the WG configuration.

The tomographic datasets recorded at the WG setup in WG configuration as well as the PB datasets used for the segmentation of the vascular system and the respective laboratory datasets were uploaded to https://doi.org/10.5281/zenodo.5658380 (*Reichardt et al., 2021*).

## Automated tissue analysis and classification of pathologies

Next, the reconstructed 3d tissue structure is analyzed by an automated workflow involving differential operators and subsequent statistical representations based on the structure tensor analysis. Instead of semantic analysis of specific structures (vessels, cardiomyocytes, ect), which is considered further below, we first target geometric properties encoded by gray value derivatives, possible prototypical distribution of these parameters in a sample, and the respective variations within and between groups. This can then later be interpreted also in view of semantic image information. A high local anisotropy and consistent orientation field, for example, can be indicative of an intact tissue with well-ordered cardiomyocyte chains. For all samples, eigenvector and eigenvalues were computed for all sampling points in the reconstructed volume. This information then includes the orientation (quasi-) vector as defined by the smallest eigenvector, as well as the shape measures for all points. As a word of caution, however, one has to keep in mind that these properties also depend on tissue preservation and preparations, as well as on the measurement and reconstruction. For this reason, the latter has to be carried out using identical workflows and parameters for all samples.

*Figure 5* shows the results of the structure tensor analysis for all samples reconstructed from LJ scans. In *Figure 5A* the mean values of the shape measures ($\mu_l, \mu_p, \mu_s$) for all datasets are plotted in a shape-measure diagram, constructed as for ternary mixtures. Sample groups are indicated by color: control-green, Covid-19-red, myocarditis-blue and influenza-yellow. Already in this plot, differences between the groups can be identified. Compared to the Ctr, the pathological groups are shifted towards lower $C_l$, indicating a less-pronounced fiber-like structure, and to higher $C_s$, reflecting a larger amount of isotropic symmetry. The Cov, Inf and Myo groups differ mainly in the $C_p$ coefficient. From Inf, to Myo and Cov, the point clouds of each group exhibit successive shifts toward increased $C_p$. However, these differences in μ are quite small, and it is not possible to classify samples only based on the average value of the shape measure. Instead, the distribution of real-space sampling points in each sample should be taken into account. *Figure 5B and C* show the area $A_\eta$ and the eccentricity $e$, respectively, of the ellipse formed by the PCA eigenvectors $\mathbf{u_1, u_2}$, for each sample, color-coded by groups. The corresponding box-whisker plots indicate a significant difference in $A_\eta$ between Cov and Ctr (Welch t-test, $p = 0.0389$) as well as a Cov and Inf (Welch t-test, $p = 0.0403$). Concerning $e$, Cov tissues differs also from Myo (Welch t-test, $p = 0.0611$). Small values of $A_\eta$, as obtained for Ctr, indicate a homogeneous tissue structure, while large values are obtained for samples with a more heterogeneous tissue composition. The parameters for each sample group are tabulated in *Table 4*. The large intra-group variance reflects the pronounced variability between individual subjects, which is in line with experience of conventional histology. The complete summary of all samples individually is given in *Appendix 1—figure 1*, *Appendix 2—table 2*. The results for the stitched tomographic datasets (PB setup) of Cov and Ctr are also shown in *Appendix 1—figure 1*.

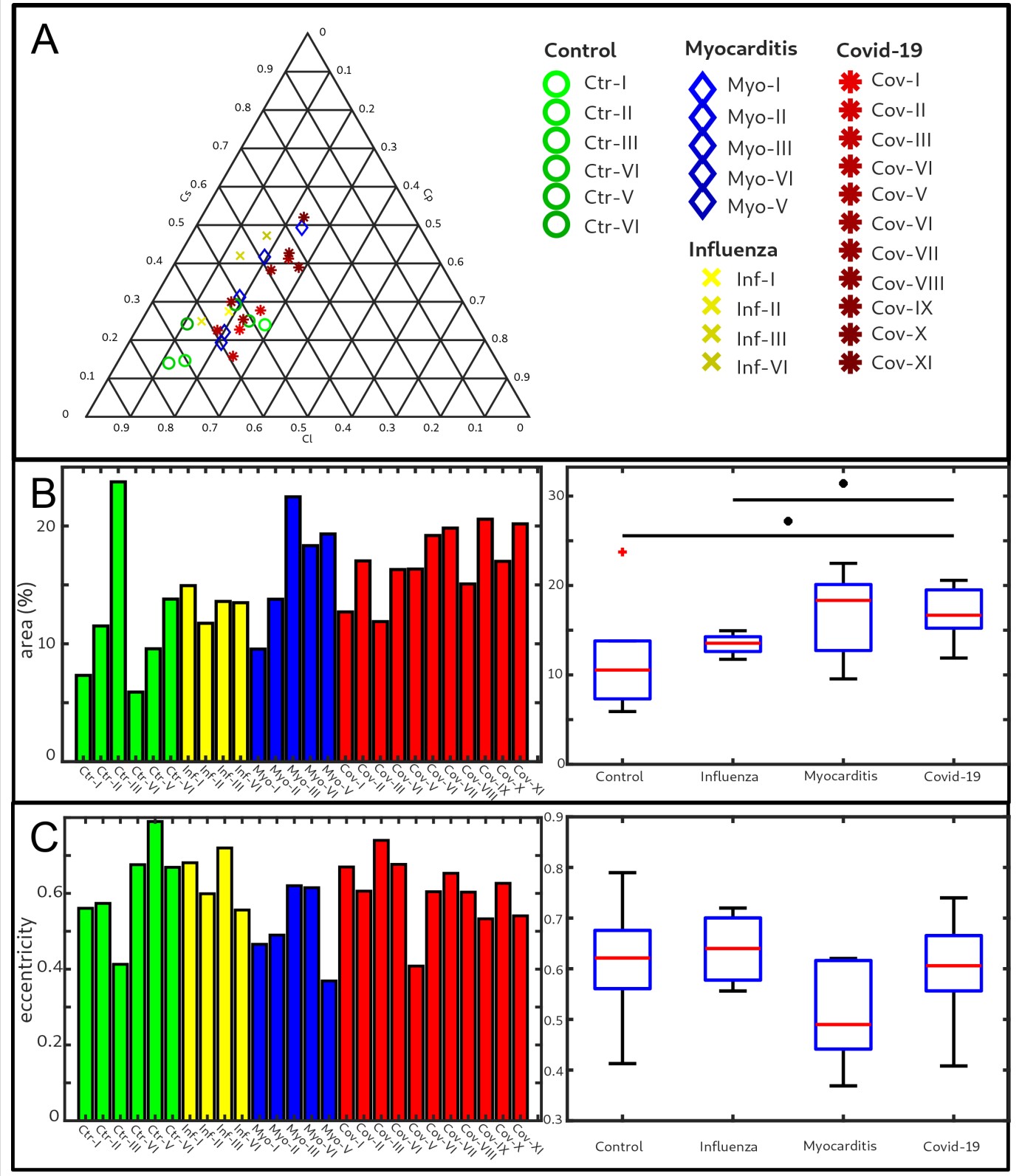

**Figure 5.** Clustering of LJ data sets. (**A**) Ternary diagram of the mean value of the shape measures for all datasets. The control samples (green) show low $C_s$ values, while samples from Covid-19 (red), influenza and myocarditis (blue) patients show a larger variance for $C_s$. (**B**) The fitted area of the elliptical fit from the PCA analysis of the shape measure distribution is an indicator for the variance in tissue structure. For Control and influenza sample this value differs significantly from the Covid-19 tissue. (**C**) The eccentricity of the fit indicates if the structural distribution in shape measure space has a preferred direction along any axis. The value of the myocarditis samples is comparable low.

**Table 4.** Parameters of the cardiac tissue obtained from LJ reconstructions.
For all sample groups the mean value and standard deviation of the mean shape measures $\overline{\mu_l}$, $\overline{\mu_p}$, $\overline{\mu_s}$ area of the elliptical fit $\overline{A_\eta}$ (%) and the eccentricity $\overline{e}$ is shown.

| Group | $\overline{\mu_l}$ | $\overline{\mu_p}$ | $\overline{\mu_s}$ | $\overline{A_\eta}$ (%) | $\overline{e}$ |
|---|---|---|---|---|---|
| Control | 0.60± 0.11 | 0.18± 0.07 | 0.22± 0.06 | 11.98± 6.42 | 0.61± 0.13 |
| Covid-19 | 0.44±0.12 | 0.23±0.03 | 0.32±0.11 | 16.92± 2.91 | 0.61± 0.09 |
| Myocarditis | 0.47±0.14 | 0.21± 0.02 | 0.33±0.13 | 16.69± 5.06 | 0.51± 0.12 |
| Influenza | 0.49±0.11 | 0.16±0.02 | 0.35±0.12 | 13.44± 1.31 | 0.63± 0.07 |

## Characterization of the vascular system

*Figure 6* reports on the segmentation and analysis of the vasculature. A surface rendering of the segmented vessels is shown in the top row, on the left for a Ctr sample (Ctr-III) and on the right for a Cov sample (Cov-IV). In Ctr, the vessels are well oriented and show a relatively constant diameter and a smooth surface. In Cov, the vessels show large deviations in diameter and the surface of the vessels is not as smooth as in Ctr. Furthermore, closed loops within the microvasculature can be identified. In *Figure 6C*, one of these vessel loops (marked with a blue line) in the Cov dataset is highlighted by a minimum intensity projection over ±30 slices around the centered slice. This pathological formation of a loop is indicative for an intermediate state in the process of IA. The corresponding vessel segmentation is depicted in *Figure 6D*, with a simplified vessel graph superimposed as black lines. Based on the simplified vessel graph, the connectivity of the capillaries can further be quantified. In total 19,893 nodes for the Cov sample and 8068 nodes in the segmentation of the Ctr were used. *Figure 6E* shows the probability density function (PDF) of the degree of connectivity $n$ for control and Covid-19 samples. It indicates a higher amount of branching points in the Covid-19 sample. This is also confirmed by the ratio of endpoints of vessels ($n = 1$) to the branching points ($n \geq 3$). Note, that the amount of nodes with $n > 3$ is almost negligible. While the Ctr data shows approximately the same number of endpoints and branching points, the Cov segmentation show almost a ratio of 1:1.5, indicating a higher degree of cross-linking or loop formation of the capillary network.

An exemplary scanning electron micrograph of a Covid-19 sample is shown in *Figure 6F*. IA was identified via the occurrence of tiny holes with a diameter of 2–5µm in SEM of microvascular corrosion casts. Capillaries display the presence of characteristic intussusceptive pillars (marked by black arrows).

## Summary, conclusion, and outlook

This is the first report of a comprehensive 3d analysis of cardiac involvement in tissue of Covid-19, influenza and coxsackie virus infections using X-ray phase-contrast tomography of human FFPE heart tissue. In summary, a high amount of distinct caliber changes of blood filled capillaries in samples of Covid-19 (Cov) patients was identified compared to the control group (Ctr) as well as to coxsackie virus myocarditis (Myo) and influenza (Inv). This can readily be explained by a much higher prevalence of micro-thrombi in Cov compared to other viral pneumoniae (e.g. influenza), as has previously been reported in Covid-19 lungs. Most importantly, high-resolution synchrotron data revealed distinct alterations of the vasculature, with larger variation in vessels diameters, intravascular pillars and amount of small holes, indicative for IA. Branching points of vessels were quantified based on graph representations, after segmentation of vessels based on deep learning. For this purpose, a network for 3d datasets (V-net) was trained with sparse annotations. In Cov, the vasculature also showed a higher degree of branching. Further, SEM data showed a high amount of holes in the capillaries, indicating the presence of multiple intussusceptive pillars as a first stage of IA. The presence of intraluminar pillars was also confirmed by the high resolution reconstruction obtained from WG acquisitions. Accordingly, we could -for the first time— visualize the presence of IA via destruction-free X-ray phase-contrast tomography not only in the heart but also for the first time in FFPE-tissue. Thus, IA is also a hallmark of Covid-19 inflammation in the heart, analogous to pulmonary previously reported for lung (*Ackermann et al., 2020b*). This finding is in line with the concept of Covid-19 as a systemic and multi-organ angiocentric entity.

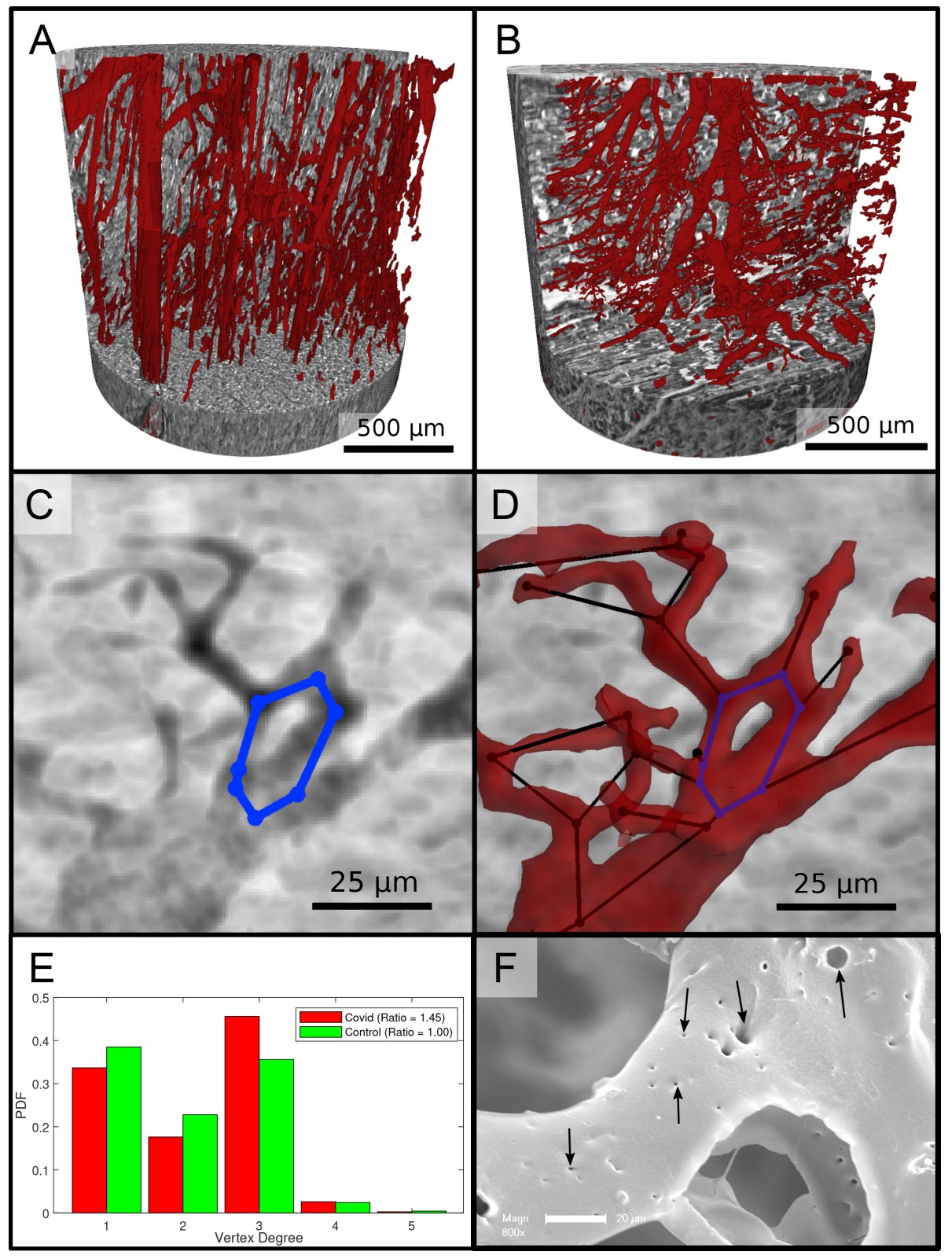

**Figure 6.** Segmentation of the vascular system in cardiac samples. (**A**) Segmentation of the vessels of a Ctr sample. The vessels are well oriented and show a relatively constant diameter. (**B**) Segmentation of the vessels of a Covid-19 sample. The vessels show large deviations in diameter and the surface of the vessels is not as smooth as in the control sample. (**C**) Filtered minimum projection of an area of the reconstructed electron density of the Cov sample to highlight a vessel loop marked in blue. (**D**) Surface rendering of the segmented vessel and vessel graph in an area of the Cov sample. Scale

*Figure 6 continued*

bars 25 $\mu$m. (**E**) Comparison of node degree $n$ between control and Covid-19. Ratio refers to the number of graph branch points ($n > 2$) divided by the number of end points ($n = 1$). (**F**) Exemplary scanning electron microscopy image of a microvascular corrosion casting from a Covid-19 sample. The black arrows mark the occurrence of some tiny holes indicating intraluminar pillars with a diameter of 2 $\mu$m to 5 $\mu$m, indicating intussusceptive angiogenesis. Magnification x800, scale bar 20 $\mu$m.

The reconstructed electron density of the Cov sample group also showed that concordant with the edema found in conventional histopathology assessment, the cardiomyocytes are not as densely packed as in the control (Ctr) group, leading to larger paraffin inclusions between the cells. Pathological alterations of the tissue architecture were further quantified in terms of non-semantic shape measures, derived from gray value differential operators, using the structure tensor approach. Since the shape measures not only depend on the tissue structure but also on the data acquisition and reconstruction parameters, the entire data acquisition and workflow was optimized and then kept constant for the entire sample series, covering the different pathologies (Cov, Inf, Myo) and control (Ctr) group samples. Importantly, this was already possible at a home-built compact laboratory $\mu$CT, based on a liquid metal jet source and optimized phase retrieval, which is important for future translation and dissemination of the methodology developed here. Fully automated PCA analysis then yielded the eigenvectors of the structure tensor at each sampling point of the reconstruction volume, and for each sample. The corresponding distributions showed significant difference in architecture between Cov from all other groups Inf, Myo or Ctr groups, and these differences could be interpreted again by inspection of the reconstruction volumes, that is reflecting for example tissue compactness, orientation of the cardiomyocytes and the degree of anisotropy.

Compared to related studies (*Walsh et al., 2021*), which focused on the analysis of entire human organs, we investigated the cardiac structure from the scale of 3.5 mm biopsy punches down to a resolution showing subcellular and supramolecular structures such as myofibrils and intussusceptive pillars.

Future improvements in segmentation and quantification will be required to fully exploit the structural data acquired here, or in similar studies. To this end, augmented image processing algorithms, deep learning, classification for example based on optimal transport, and the consolidation of the above in form of specialized software packages has to be considered. Technical improvements towards higher resolution and throughput can also be foreseen. Already at present, parallel beam synchrotron data acquisition (GINIX endstation, P10 beamline of PETRA III/DESY) completes a biopsy punch tomogram within 1.5 min, at a pixel size of 650 nm, and a volume throughput of $10^7$ $\mu$m$^3$/s. Importantly, the image resolution and quality is sufficient to segment vasculature and cytoarchitectural features of interest, also and especially for standard unstained paraffin-embedded tissue used in routine diagnostics. The data acquisition rate and dwell time in the range of 10 ms to 20 ms (per projection) is dictated by detector readout, motor synchronisation, and data flow rather than by photon flux density for the PB setup. This is also underlined by the fact that (single-crystal) attenuators had to be used to prevent detector saturation. The situation is entirely different, however, for the waveguide cone beam setup, where the lower waveguide exit flux density, which comes with the significantly higher coherence and resolution, requires acquisition times of 200 ms to 2500 ms. Here, the projected source upgrade foreseen for PETRA IV will provide a significant gain in resolution and throughput. Robotic sample exchange will therefore be required, as well as a serious upscaling of the data management and online reconstruction pipeline. First reconstructions of heart biopsies exploiting the enhanced coherence and resolution of a waveguide holo-tomography setup already indicate that this is a very promising direction. With our presented workflow, especially in view of the laboratory system, we have for the first time implemented destruction free analysis of the ubiquitous FFPE embedded tissue readily available in every pathology lab around the world, based on an automated structure tensor and shape measures. This represents a first and major step in unlocking the extensive international FFPE archives for sub-light-microscope resolution destruction-free 3d-tissue analysis, unfolding manifold future research possibilities in human diseases far beyond Covid-19. This approach has been successfully used to classify the distinct changes in the myocardial cytoarchitecture induced by Covid-19. More importantly still, we have provided first proof for the suspected presence of IA in cardiac Covid-19 involvement, putting forward morphological evidence of a so far imprecisely defined clinical entity of great importance.

## Acknowledgements

We thank Ove Hansen for help with deep learning, Markus Osterhoff, Michel Sprung, and Fabian Westermeier for support at P10. Florian LÃ¤nger for helpful discussion, Patrick Zardo for providing control specimen, and Bastian Hartmann, Jan Goemann, Regina Engelhardt, Anette MÃ¼ller-Brechlin and Christina Petzold for their excellent technical help. It is also acknowledge DESY photon science management for the Covid-19 beamtime call and the granted beamtime. Funding This research was supported by the Max Planck School Matter to Life supported by the German Federal Ministry of Education and Research (BMBF) in collaboration with the Max Planck Society (MR,TS), as well as BMBF grant No. 05K19MG2 (TS), German Research Foundation (DFG) under Germanys Excellence Strategy -EXC 2067/1–390729940 (TS), the European Research Council Consolidator Grant XHale, 771,883 (DJ) and KFO311 (project Z2) of the DFG (DJ). Participation of PMJ was supported by a HALOS exchange stipend.

## Additional information

### Funding

| Funder | Grant reference number | Author |
| --- | --- | --- |
| Bundesministerium für Bildung und Forschung | Max Planck School Matter to Life | Marius Reichardt Tim Salditt |
| Bundesministerium für Bildung und Forschung | 05K19MG2 | Tim Salditt |
| Deutsche Forschungsgemeinschaft | EXC 2067/1-390729940 | Tim Salditt |
| H2020 European Research Council | XHale | Danny Jonigk |
| Deutsche Forschungsgemeinschaft | KFO311 (project Z2) | Danny Jonigk |
| Hanseatic League of Science | | Patrick Moller Jensen |
| H2020 European Research Council | 771883 | Danny Jonigk |

The funders had no role in study design, data collection and interpretation, or the decision to submit the work for publication.

### Author contributions

Marius Reichardt, Conceptualization, Investigation, Methodology, Software, Visualization, Writing - original draft, Writing – review and editing; Patrick Moller Jensen, Investigation, Methodology, Resources, Software, Visualization, Writing - original draft, Writing – review and editing; Vedrana Andersen Dahl, Anders Bjorholm Dahl, Methodology, Resources, Supervision, Writing – review and editing; Maximilian Ackermann, Investigation, Methodology; Harshit Shah, Investigation; Florian Länger, Investigation, Resources; Christopher Werlein, Investigation, Methodology, Writing – review and editing; Mark P Kuehnel, Investigation, Methodology, Resources, Writing – review and editing; Danny Jonigk, Conceptualization, Funding acquisition, Investigation, Methodology, Project administration, Resources, Supervision, Writing – review and editing; Tim Salditt, Conceptualization, Funding acquisition, Investigation, Methodology, Project administration, Resources, Supervision, Writing - original draft, Writing – review and editing

### Author ORCIDs

Maximilian Ackermann http://orcid.org/0000-0001-9996-2477
Christopher Werlein http://orcid.org/0000-0002-7694-4257
Tim Salditt http://orcid.org/0000-0003-4636-0813

## Ethics

Human subjects: Formalin-fixed paraffin-embedded tissue blocks of control hearts, influenza and coxsackie virus myocarditis hearts were retrieved from archived material from the Institute of Pathology at Hannover Medical School in accordance with the local ethics committee (ethics vote number: 1741-2013 and 2893-2015). Formalin-fixed paraffin-embedded tissue blocks of COVID-19 autopsy cases were retrieved after written consent in accordance with the local ethics committee at Hannover medical school (ethics vote number: 9022 BO K 2020).

## Decision letter and Author response
Decision letter https://doi.org/10.7554/eLife.71359.sa1
Author response https://doi.org/10.7554/eLife.71359.sa2

## Additional files

### Supplementary files
• Transparent reporting form

### Data availability
The tomographic datasets recorded in WG configuration as well as the PB datasets used for the segmentation of the vascular system and the respective laboratory datasets were uploaded to https://doi.org/10.5281/zenodo.4905971. Additional data (raw data, PB and laboratory reconstructions, structure tensor analysis) is curated here at University of Göttingen and at DESY can be obtained upon request from the corresponding author (tsaldit@gwdg.de); due to the extremely large size >15TB it cannot presently be uploaded easily to a public repository. The implementation of the structure tensor analysis is provided in https://lab.compute.dtu.dk/patmjen/structure-tensor. The neural network code used for the segmentation of the vasculature was uploaded to GitHub (https://github.com/patmjen/blood-vessel-segmentation copy archived at https://archive.softwareheritage.org/swh:1:rev:783df24c3068e35f2ae994cab095b4318c755b29).

The following dataset was generated:

| Author(s) | Year | Dataset title | Dataset URL | Database and Identifier |
|---|---|---|---|---|
| Reichardt M, Jensen PM, Dahl VA, Dahl AB, Ackermann M, Shah S, Länger F, Werlein C, Kühnel M, Jonigk D, Salditt T | 2021 | 3D virtual Histopathology of Cardiac Tissue from Covid-19 Patients based on Phase-Contrast X-ray Tomography | https://doi.org/10.5281/zenodo.4905971 | Zenodo, 10.5281/zenodo.4905971 |

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

## Appendix 1

### Supplementary figures

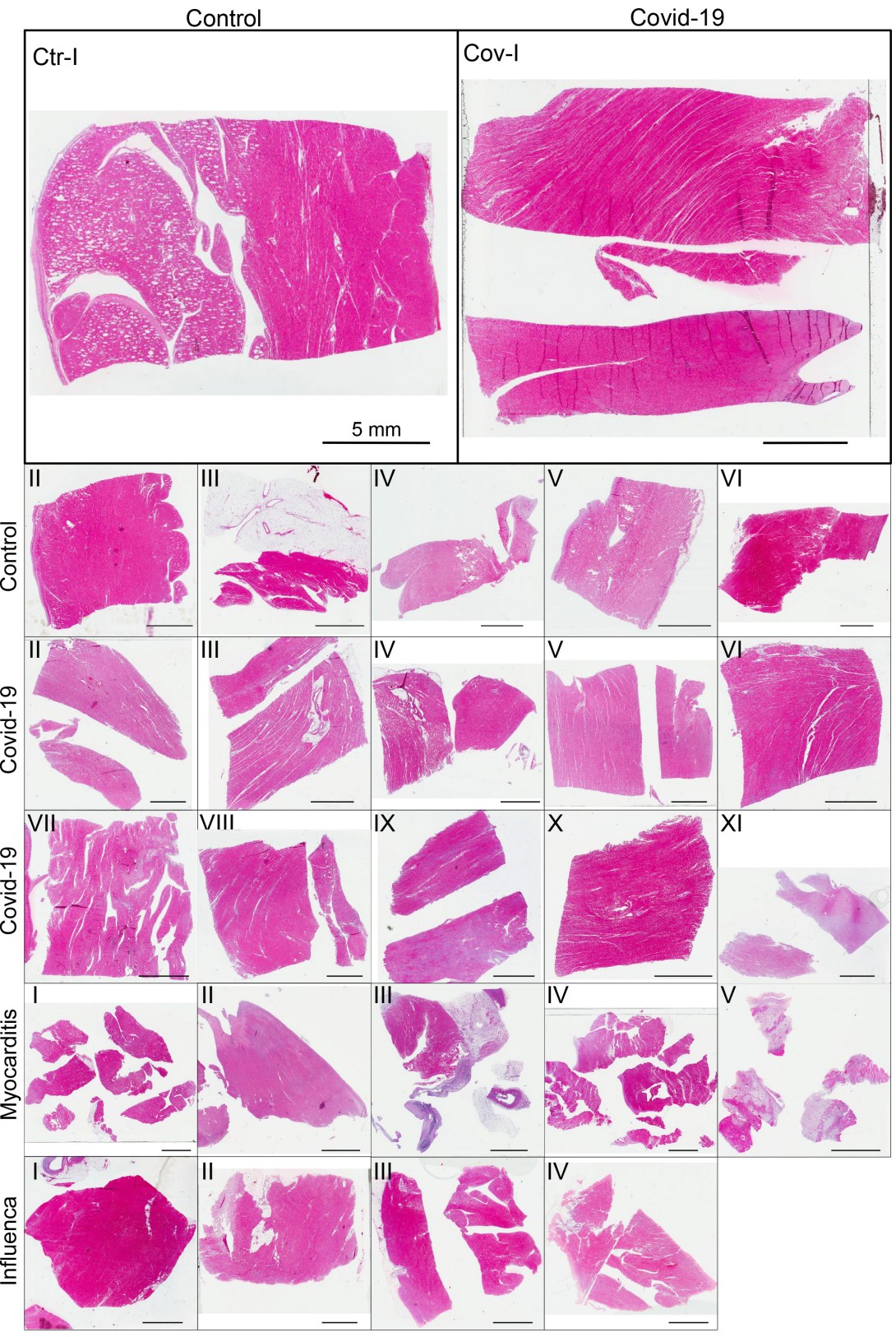

**Appendix 1—figure 1.** HE stain of all cardiac samples . Scale bar: 5 mm.

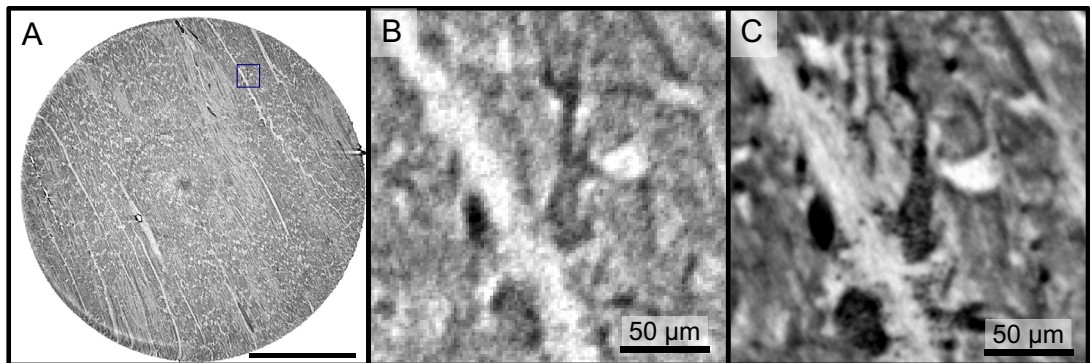

**Appendix 1—figure 2.** Reconstructions of the LJ compared to the PB setup. Comparison of the data quality of laboratory and synchrotron measurements. (**A**) slice of a laboratory reconstruction at a voxelsize of $2\,\mu$m. A region of interest containing a branching vessel is marked by a blue box which is shown in (**B**). The same area cropped from a tomographic reconstruction at the PB setup at a voxelsize of $650\,$nm is shown in (**C**). The smaller voxelsize, higher contrast and SNR of the PB scans is necessary to segment the vascular system. Scale bars: (**A**) $1\,$mm, (**B,C**).$50\,\mu$m

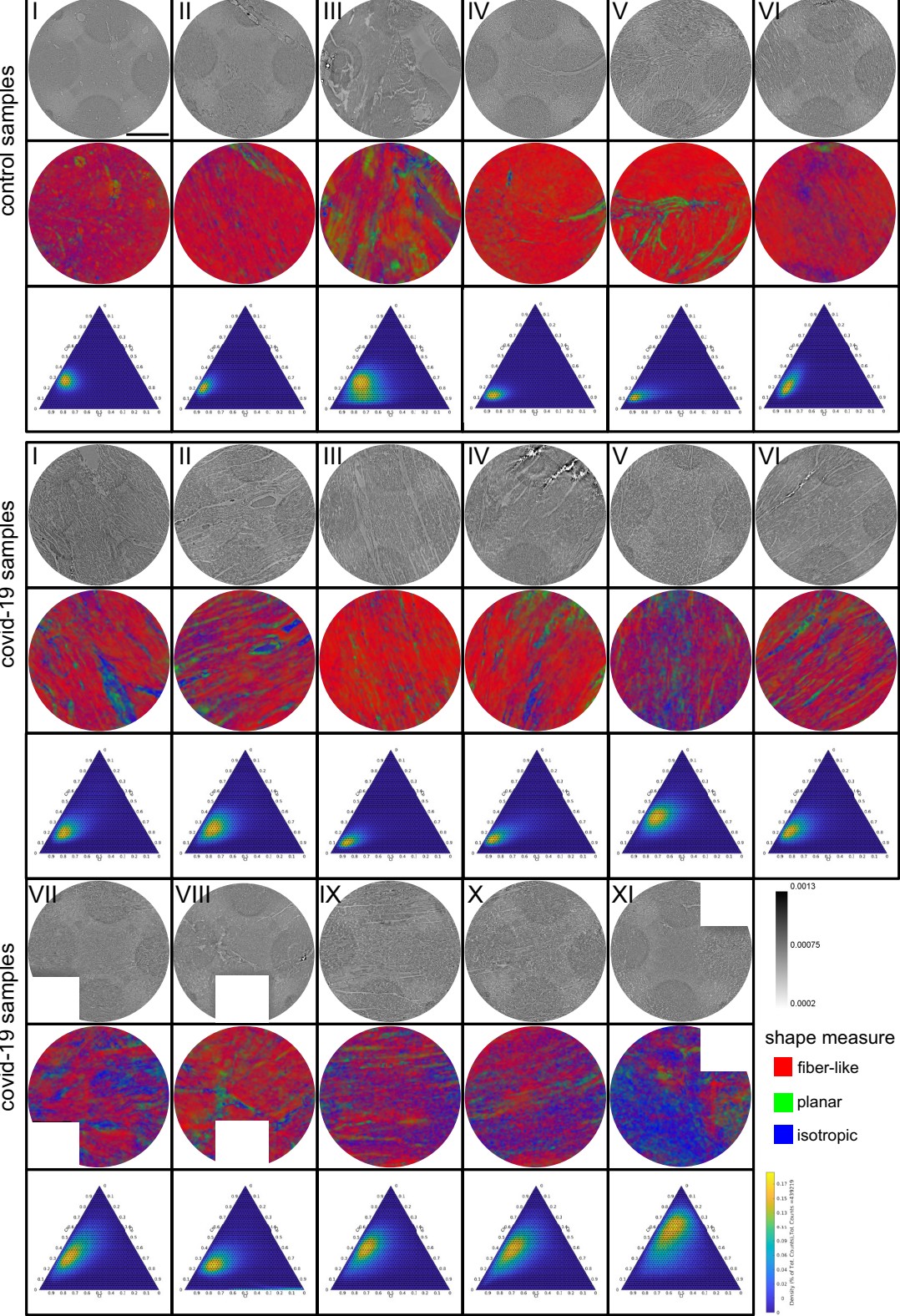

**Appendix 1—figure 3.** Shape measure of all Covid-19 and control samples reconstructed from PB data. Slices of the reconstructed electron density (stitched volumes of 3 ×three tomographic reconstructions), the corresponding slice of the shape measure and the ternary plot of the shape distribution in the entire volume are shown. Corrupted datasets were excluded from the analysis and masked in white. Scale bar: 1 mm.

## Appendix 2

## Supplementary Information: Medical background and datasets
Medical Information

**Appendix 2—table 1.** Sample and medical information.
Age and sex, clinical presentation with hospitalization and treatment. RF: respiratory failure, CRF: cardiorespiratory failure, MOF: multi-organ failure, V: ventilation, S: Smoker, D: Diabetes TypeII, H: Hypertension, I: imunsupression

| Sample no. | Age, sex | Hospitalization (days), clinical, radiological and histological characteristics |
|---|---|---|
| Cov-I | 86,M | 5d, RF, D, H, I |
| Cov-II | 96,M | 3d, RF, H |
| Cov-III | 78,M | 3d, CRF, V, D, S, H |
| Cov-IV | 66,M | 9d, RF, V, S, H |
| Cov-V | 74,M | 3d, RF, D, S, H |
| Cov-VI | 81,F | 4d, RF, S, H |
| Cov-VII | 71,M | 0d, V |
| Cov-VIII | 88,M | 2d, V, H, I |
| Cov-IX | 85,M | 5d, V, S, H |
| Cov-X | 58,M | 7d, V, H |
| Cov-XI | 54,M | 15d, V |
| Ctr-I to Ctr-III | 26, F | - |
| Ctr-IV to Ctr-VI | 36, F | - |
| Myo-I | 57,M | V, H |
| Myo-II | 23,M | |
| Myo-III | 59,M | S, H, D |
| Myo-IV | 50,M | V, S, D |
| Myo-V | 25,F | |
| Inf-I | 74,M | 9d, CRF into MOF, V, S, H |
| Inf-II | 66,F | 17d, MOF, V, H |
| Inf-III | 56,M | 3d, CRF into MOF, V |
| Inf-IV | 55,M | 24d, RF into MOF, V, S |

Structural analysis

**Appendix 2—table 2.** Parameters of the cardiac tissue (laboratory data).

| Sample | Mean (Cl,Cp, Cs) | Fitted area | Eccentricity |
|---|---|---|---|
| Ctr-I | (0.6508, 0.1069, 0.2423 ) | 7.3194 | 0.5607 |

*Appendix 2—table 2 Continued on next page*

*Appendix 2—table 2 Continued*

| Sample | Mean (Cl,Cp, Cs) | | Fitted area | Eccentricity |
|---|---|---|---|---|
| Ctr-II | ( 0.5167, 0.1907, 0.2926 ) | | 11.5130 | 0.5736 |
| Ctr-III | ( 0.5074, 0.2427, 0.2499) | | 23.7443 | 0.4128 |
| Ctr-IV | ( 0.7434, 0.1166, 0.1400 ) | | 5.9026 | 0.6757 |
| Ctr-V | ( 0.7038, 0.1495, 0.1467 ) | | 9.5763 | 0.7896 |
| Ctr-VI | ( 0.4765, 0.2835, 0.2400 ) | | 13.7973 | 0.6688 |
| mean | (0.60 ± 0.11. 0.18 ± 0.07, 0.22 ± 0.06) | | 11.98 ± 6.42 | 0.61 ± 0.13 |
| Cov-I | ( 0.5398, 0.2327, 0.2275) | | 12.7052 | 0.6696 |
| Cov-II | ( 0.4676, 0.2550, 0.2774 ) | | 17.0347 | 0.6059 |
| Cov-III | ( 0.5896, 0.2526, 0.1578) | | 11.8845 | 0.7399 |
| Cov-IV | ( 0.5911, 0.1833, 0.2255 ) | | 16.3040 | 0.6765 |
| Cov-V | ( 0.3371, 0.2505, 0.4124) | | 16.3445 | 0.4081 |
| Cov-VI | ( 0.5184, 0.2279, 0.2537) | | 19.1954 | 0.6044 |
| Cov-VII | (0.3912, 0.2262, 0.3826) | | 19.8206 | 0.6530 |
| Cov-VIII | ( 0.5227, 0.1776, 0.2997) | | 15.0791 | 0.6033 |
| Cov-IV | (0.3253, 0.2851, 0.3897 ) | | 20.5768 | 0.5329 |
| Cov-X | (0.3283, 0.2446, 0.4271 ) | | 16.9989 | 0.6266 |
| Cov-XI | ( 0.2484, 0.2314, 0.5202 ) | | 20.1815 | 0.5407 |
| mean | $(0.44 \pm 0.12, 0.23 \pm 0.03, 0.32 \pm 0.11)$ | | 16.92 ± 2.91 | 0.61 ± 0.09 |
| Myo-I | (0.5777, 0.2018, 0.2206 ) | | 9.5528 | 0.4656 |
| Myo-II | (0.3887, 0.1943, 0.4170 ) | | 13.7853 | 0.4899 |
| Myo-III | ( 0.5984, 0.2081, 0.1935 ) | | 22.4768 | 0.6202 |
| Myo-IV | ( 0.4974, 0.1908, 0.3117 ) | | 18.3306? | 0.6149 |
| Myo-V | (0.2664, 0.2402, 0.4933 ) | | 19.3212 | 0.3689 |
| mean | (0.27 ± 0.14, 0.24 ± 0.02, 0.49 ± 0.13) | | 16.69 ± 5.06 | 0.51 ± 0.12 |

*Appendix 2—table 2 Continued*

| Sample | Mean (Cl,Cp, Cs) | Fitted area | Eccentricity |
|---|---|---|---|
| Inf-I | (0.3561, 0.1714, 0.4724 ) | 14.9393 | 0.6808 |
| Inf-II | ( 0.4423, 0.1376, 0.4201 ) | 11.7445? | 0.5991 |
| Inf-III | ( 0.6150, 0.1361, 0.2489 ) | 13.5988 | 0.7198 |
| Inf-IV | ( 0.5404, 0.1849, 0.2747 ) | 13.4885 | 0.5561 |
| mean | (0.49 ± 0.11, 0.16 ± 0.02, 0.35 ± 0.11) | 13.44 ± 1.31 | 0.63 ± 0.07 |

## Datasets

The tomographic datasets recorded at thein WG setup as well as the PB datasets used for the segmentation of the vascular system were uploaded to https://doi.org/10.5281/zenodo.5658380.

