## [Editor Report]

In this manuscript the authors demonstrate that X-ray imaging delivers more detailed information than standard histology by analyzing 3D information in myocardial tissue obtained from COVID-19 patients. The findings are of particular interest regarding the segmentation of the vascular network and intussusceptive angiogenesis. The authors introduce the utilization of machine learning, and state-of-the-art techniques of X-ray phase contrast which is likely to advance future work in this field. Finally, with this manuscript the authors also provide new, more detailed insights into the pathologies associated with cardiac injury due to COVID-19.

---

## [Decision Letter]

**Decision letter after peer review:**

Thank you for submitting your article "3D virtual Histopathology of Cardiac Tissue from Covid-19 Patients based on Phase-Contrast X-ray Tomography" for consideration by *eLife*. Your article has been reviewed by 2 peer reviewers, and the evaluation has been overseen by a Reviewing Guest Editor and a Senior Editor. The reviewers have opted to remain anonymous.

The reviewers have discussed their reviews with one another, and the Reviewing Guest Editor has drafted this to help you prepare a revised submission.

Essential revisions:

Please refer to revisions requested by the reviewers below, particularly as outlined by Reviewer 1 whose critique will greatly help to strengthen this manuscript and its conclusions.

*Reviewer #1 (Recommendations for the authors):*

Below the authors will find my additional comments to increase the clarity of their work and making it suitable for *eLife*.

Abstract: "first time" Several times the authors mention this paper that this is the first time that such a study is done. It's only partially true. Over the past two years a lot of study have been published or are currently in accepted state about analysis of COVID-19 cardiac samples (for instance Walsh 2021 that the authors cite analyze partially whole heart as well).

l69: "we have introduce". The authors are not the only group working on this topic. A more general sentence saying that this field is growing would be more appropriate.

l76: "entire organ" -> "entire human organ". The example cited of Walsh is on human organs. However, a lot of work has been already done in the past in entire organ of animals such as mouse.

l79: "cytoarchitecture". It's a bit misleading as we expect to see results of the level of TEM. It's true that this level is reached for the results presented with the WG configuration but partially for the core analysis of the paper. Indeed the results focus on the general organization and vasculature.

Why the analysis has not been done in parallel or at the same time?

Is there any risk for this technique on future analysis due to sample degradation for instance?

l92: "based on visual impression" -> Sounds not scientific. Have the data been analyzed by a pathologist for assessment? Is that how it is done by pathologist?

l93: "automated image processing". Partially true, one still need to spend some times for doing manual segmentation. In the document nothing is mentioned about the quantity of images necessary for this process neither.

Figure 1: Later in the paper, the authors are describing the 3 methods used for the analysis as "LJ setup", "PB configuration", "WG configuration". First for clarity I would choose either setup either configuration. Then, I would also introduce those names in the figure or at least in the legend.

(B) -> "from one of the control and one of the Covid-19 samples". Otherwise it's misleading and difficult to understand that only one of each has been analyzed. How the choice actually has been made between the samples?

(D) plane of 3x3. Why 3x3 and not larger or smaller? Would a 360 or half scans be able to cover the interesting part of the biopsy that is in the end analyzes ? (i.e. to avoid the cropping due to the holder)?

(E) "was taken from a control sample". The analysis have been done on a COVID-19 sample as well, no?

Appendix Figure 1: The figure of the Haematoxylin and Eosin staining presents images too pink and no details of the microstructures can be really seen. Are those slices corresponding exactly to the samples before the biopsy punch? Have they been compared to slices for all samples on the X-ray datasets?

l113: "Biopsy punches". How the areas have been selected? Why the amount of biopsy punch are not identical for all samples? i.e. why not taking the entire block or selecting 2 samples per patients as for the CTRL?

l117: "one (Ctr) biopsy". One control and one COVID-19 are presented.

l123: It would be nice to have the source size of the liquid jet as well as the corresponding magnification factor. Where the broad spectrum used or only 9.25 keV? What is the true resolution compare to the pixel size?

l132: The measurement have been done at a different energy. What is the impact of such changes on the results? Would a higher energy be interesting?

l136: "the continuous scan mode". This is the property of the rotation stage I guess, and not the reason of being able to perform 3x3 tomography scans. Isn't it more a question of speed and stability of the stages? Have the scan been perform over 180 degrees?

l137: "dark field images were taken" I guess that flat field as well?

l138: "150 tomographic scans were recorded". It's not clear that this is for the entire amount of samples (with 3x3 for each).

l141: "1mm diameter biopsy punch". How this has been performed? Seems very tricky to extract 1mm rod from the 3.5mm one. How the area has been chosen. Was the height of the sample the same?

l139-153. It was not clear for me from the beginning that the 2 samples have been analyzed in 2 different configuration of the setup. A sentence introducing and explaining this would be helpful.

Table2: the source sample distance is missing for the PB configuration. Why the number of projections in the case of the PB configuration is the double than the 2 other configuration. Why the amount of empties/dark field so large? Empties is not the common name used in the field. Usually the term "flats" is preferred.

What means the acquisition time 3x0.6?

It would be interesting to give the total scan time for each technique.

l156: "local median filtering"

As the authors are using different phase contrast techniques for their analysis, a short introduction and a clearer organization of this paragraph would help in the comprehension for non-specialist. Some references could also be added. It is not easy to follow the difference between the reconstruction and phase retrieval techniques used for each technique.

l177: "datasets were binned by a factor of 2".

Tomographic datasets can indeed become very heavy, specially after stitching. However, here the purpose of using synchrotron is to reach higher resolution and higher throughput than in laboratory. When binning the data, the pixel size / resolution is also reduced. Therefore leading to a similar pixel size than in laboratory. Have the analysis be computed on the binned data or on the full datasets?

Table3: 1/ The d/b ratio values are real values of δ/β or just the result of the ratio? 2/ One can not understand what are the parameters.

l183: "corrupted datasets" What was the issues here? Need clarifications.

l197: "32 pixels for PB datasets" If the analysis has been done on binned data, it would lead to different size compare to the 12 pixels for the LJ acquisitions?

"A smoothing parameter of 2 pixels" It would be nice to have more explanation here. Have this been applied to both PB and LJ datasets?

l213: "the paraffin surrounding" Why the mask has been applied only on the LJ datasets. Isn't it included as well in the acquisition of the PB datasets?

l214: "Since one axis of the shape measure is redundant" Could the author clarify this point?

l231: "A small number of axis-aligned 2D slices was annotated". It would be very interesting to know the amount of annotated datasets that have been necessary to perform the analysis. Indeed, this is the most critical point when performing machine learning technique. Sparsely annotated data sets is a very promising technique, specially for analysing large amount of images. What was the percentage of images of annotated volumes were kept for training / validation?

l236: Why 96x96x96 voxels? What was the total size of the datasets?

l239 "256 subvolumes" How is this amount representative of the entire datasets?

l243: "A separate model" why was it necessary? Doesn't it create bias in the results?

l247: "adding additional annotations" Do I understand correctly that the analysis is run a first time, then the authors look at the images and visually correct some of them and re-run the analysis?

l314: "different areas of the same heart" How those areas have been chosen. Why 3 samples per patients?

l323: "dark stripes" In the figure it's easy to distinguish white stripes, but not the dark ones. Maybe small areas or markers would help to follow the description.

l329 / Figure 3: How to make the difference between paraffin cracks and inclusion compare to other features?

l337: "samples near an artery". How the areas have been chosen and what is the impact on the results presented?

l360: Appendix 1 Figure 2: would be nice to have arrows in the figure to understand what is erythrocytes and capillaries.

l368: "340x340x340um3": Why this size has been chosen and not the entire volume?

l371: "cytosol" would be nice to have this indicated in the figure.

l375: "nucleus" It seems that we can see only on in the figure. Do we see others in the 3D volume?

l386: "speeding-up the measurement sequence" It would be interesting to indeed grasp the difficulties of such measurement and time necessary.

Table 4: mean shape mu_p and mu_s are of the same order (ctrl or diseased) for mu_l there is a difference between control versus diseased however the σ is pretty large (almost 30% in certain case). Similar remarks for the elliptical fit (54% for σ in the case of Control…). The Authors are aware and found an interesting way of representing the results (Figure 5A). However, can the authors comments on why the errors are so huge, and how can one conclude in term of differences according to those results? Because even on the graph, we see tendency but to make strong concluding statement is difficult. I would therefore modify part of the eluding sentence in this direction.

Maybe could they compare with histology analysis conclude beforehand to help in the understanding.

l403: Authors are aware that their methods have some limitation. For instance "depend on tissue preservation and preparations".

l419: Welch t-test: Would be good to have a reference an explanation sentence.

l427: "Surface rendering" from which acquisition? Would be nice also to refer to the corresponding method paragraph.

l438: Could the authors explains a bit more what is the probability density function and how it is obtained. Figure 6E: The authors state clearly that their is a higher amount of branching points in the Covid19 sample. However, this is the results on one sample and the figure is not clearly stating that. For instance for 5 vertex degree the control seems superior than the covid.

l449: "first report" What about Walsch 2021?

l465: "non-destructively". It's not totally non-destructive technique as one still need to make a biopsy punch on the blocks.

l470: "conventional histopathology assessment" It would be nice to have a reference on the histology conventional analysis are adding in supplementary material correlation between X-ray images and histology images.

Figure 5: how those 2 samples have been chosen compare to the previous selection.

l493: "volume throughput of 10ˆ7 um3 /s. Maybe a simpler presentation would be more meaningful for the general audience, like acquisition time for one sample with which setup.

l497: rather than by photon flux: This is valid for the synchrotron acquisition I guess. But then what would be the purpose of the new synchrotron source if the flux is already not fully exploited? Same question concerning the following sentence about the attenuators used to prevent detector saturation. What about dose on the sample then?

l500: why so huge acquisition time range 200ms to 2500ms?

Appendix 1 Figure 3: In all the images we see like a cross in the images creating a white blur. Can the authors comment on that? Specially because grey levels are used for the analysis. How this effect can affect the results? A comment should also be added for the square missing (i.e. areas for corrupted files). Why measurements have not be redone locally?

*Reviewer #2 (Recommendations for the authors):*

It would be important to improve the samples' statistics.

---

## [Author Response]

Reviewer #1 (Recommendations for the authors):Below the authors will find my additional comments to increase the clarity of their work and making it suitable for eLife.Abstract: "first time" Several times the authors mention this paper that this is the first time that such a study is done. It's only partially true. Over the past two years a lot of study have been published or are currently in accepted state about analysis of COVID-19 cardiac samples (for instance Walsh 2021 that the authors cite analyze partially whole heart as well).

Indeed there are multiple studies on structural changes of cardiac tissue due to Covid-19. However, this manuscript presents the first three-dimensional structural comparison of cardiac tissue from Covid-19 patients to different diseases. To avoid confusions we removed “first time” from the abstract.

l69: "we have introduce". The authors are not the only group working on this topic. A more general sentence saying that this field is growing would be more appropriate.

We have rephrased the sentence accordingly.

l76: "entire organ" -> "entire human organ". The example cited of Walsh is on human organs. However, a lot of work has been already done in the past in entire organ of animals such as mouse.

We have rephrased the sentence.

l79: "cytoarchitecture". It's a bit misleading as we expect to see results of the level of TEM. It's true that this level is reached for the results presented with the WG configuration but partially for the core analysis of the paper. Indeed the results focus on the general organization and vasculature.

We have rephrased “cytoarchitecture” to avoid confusions.

Why the analysis has not been done in parallel or at the same time?Is there any risk for this technique on future analysis due to sample degradation for instance?

All samples have been investigated during the clinical routine and were provided for X-ray analysis. Formalin fixed paraffin embedded tissue are stable and do not degrade over time. At the relatively low dose invested, we have no indications of any sample structural degradation during or by the X-ray scans. However, we first wanted a clinical more established assessment before the X-ray analysis.

l92: "based on visual impression" -> Sounds not scientific. Have the data been analyzed by a pathologist for assessment? Is that how it is done by pathologist?

The data as well as the histology shown in the appendix was analyzed by pathologists. We have rephrased the sentence.

l93: "automated image processing". Partially true, one still need to spend some times for doing manual segmentation. In the document nothing is mentioned about the quantity of images necessary for this process neither.

The analysis of the laboratory data was performed automatically as described in the “Structure tensor analysis” section. The V-net based segmentation required manual annotation which acts as ground truth for the training of the network, but also here the statistical processing of the results based on graph theory was performed automated.

Figure 1: Later in the paper, the authors are describing the 3 methods used for the analysis as "LJ setup", "PB configuration", "WG configuration". First for clarity I would choose either setup either configuration. Then, I would also introduce those names in the figure or at least in the legend.

The wording has been changed to “setup” throughout, and is mentioned in the legends.

(B) -> "from one of the control and one of the Covid-19 samples". Otherwise it's misleading and difficult to understand that only one of each has been analyzed. How the choice actually has been made between the samples?(D) plane of 3x3. Why 3x3 and not larger or smaller? Would a 360 or half scans be able to cover the interesting part of the biopsy that is in the end analyzes ? (i.e. to avoid the cropping due to the holder)?(E) "was taken from a control sample". The analysis have been done on a COVID-19 sample as well, no?

We have now rephrased the caption to clarify the experimental setup and process of sample preparation and data acquisition.

Appendix Figure 1: The figure of the Haematoxylin and Eosin staining presents images too pink and no details of the microstructures can be really seen. Are those slices corresponding exactly to the samples before the biopsy punch? Have they been compared to slices for all samples on the X-ray datasets?

The micrographs shown in Appendix Figure 1 were recorded from all samples before a biopsy punch was taken, thus they correspond to the upper slice of the tomographic reconstructions. The presented H and E images represent the complete FFPE-sample. Both, histology and phase-contrast tomography data, were investigated by pathologists to describe the cardiac structure.

l113: "Biopsy punches". How the areas have been selected? Why the amount of biopsy punch are not identical for all samples? i.e. why not taking the entire block or selecting 2 samples per patients as for the CTRL?

The 3.5mm punches were selected to contain as much tissue as possible. We took one biopsy from each pathological tissue block which could be provided for this study. Fortunately, multiple control tissue blocks from the same donor could be provided, thus we were able to take multiple biopsies from the same patient.

l117: "one (Ctr) biopsy". One control and one COVID-19 are presented.

Indeed! We included the missing information in the text.

l123: It would be nice to have the source size of the liquid jet as well as the corresponding magnification factor. Where the broad spectrum used or only 9.25 keV? What is the true resolution compare to the pixel size?

The definition of the geometric magnification is provided in the text. We now also have included the corresponding magnifications in Table 2. For the phase-retrieval we have used the value of the Kα line. The resolution was not determined for this experiment, but we could show that the resolution is in the range of less than 2 px for this configuration in a different work. We now have included the respective reference (Reichardt et al., JMI, 2020).

l132: The measurement have been done at a different energy. What is the impact of such changes on the results? Would a higher energy be interesting?

In general, the interaction of X-ray radiation with matter is lower at higher energies. Moreover, the photon energy has to be optimized also with respect to the X-ray optics as undulator, monochromator, focusing optics and waveguide channel. Hence, different setups enable different photon energies, and optimizing these individually (for each setup) results in higher image quality.

l136: "the continuous scan mode". This is the property of the rotation stage I guess, and not the reason of being able to perform 3x3 tomography scans. Isn't it more a question of speed and stability of the stages? Have the scan been perform over 180 degrees?

That is correct. We have rephrased the paragraph and included the information on the angular range (360°!).

l137: "dark field images were taken" I guess that flat field as well?

Yes. The information was added.

l138: "150 tomographic scans were recorded". It's not clear that this is for the entire amount of samples (with 3x3 for each).

We acquired 3x3 datasets for 17 samples (153 tomograms), as we now explain in more detail.

l141: "1mm diameter biopsy punch". How this has been performed? Seems very tricky to extract 1mm rod from the 3.5mm one. How the area has been chosen. Was the height of the sample the same?

Indeed it takes a firm hand to take a 1mm biopsy from the 3.5 mm core. Thus, the small punch was extracted from the center of the 3.5mm biopsy punch. The height of both biopsies is the same.

l139-153. It was not clear for me from the beginning that the 2 samples have been analyzed in 2 different configuration of the setup. A sentence introducing and explaining this would be helpful.

The process of sample preparation and data acquisition is described in line 113-119

Table2: the source sample distance is missing for the PB configuration.

The source to sample distance was 88 m. We added the information to the table.

Why the number of projections in the case of the PB configuration is the double than the 2 other configuration. Why the amount of empties/dark field so large?

Since the data acquisition in the PB configuration can be performed in less than 2 minutes we extended the number of projections and flat field images.

Empties is not the common name used in the field. Usually the term "flats" is preferred.

We changes the name in the entire manuscript to flat field images.

What means the acquisition time 3x0.6?

Since the detector chip saturates at longer exposure times, we averaged 3 projections with an acquisition time of 0.6s. We now included a detailed description of the acquisition process.

It would be interesting to give the total scan time for each technique.

We have included the time for the different configurations.

l156: "local median filtering"As the authors are using different phase contrast techniques for their analysis, a short introduction and a clearer organization of this paragraph would help in the comprehension for non-specialist. Some references could also be added. It is not easy to follow the difference between the reconstruction and phase retrieval techniques used for each technique.

We have restructured the paragraph, added some general information and included the reference for the toolbox used for the analysis at the beginning of the section.

l177: "datasets were binned by a factor of 2".Tomographic datasets can indeed become very heavy, specially after stitching. However, here the purpose of using synchrotron is to reach higher resolution and higher throughput than in laboratory. When binning the data, the pixel size / resolution is also reduced. Therefore leading to a similar pixel size than in laboratory. Have the analysis be computed on the binned data or on the full datasets?

The analysis has been performed on the binned data. The difference in data quality is shown in Appendix Figure 2.

Table3: 1/ The d/b ratio values are real values of δ/β or just the result of the ratio? 2/ One can not understand what are the parameters.

The δ/β-ratio given in Tab.3 is the ratio which was used for phase retrieval (see Cloetens et al., 1999). In other words, they have to be regarded as effective parameters (guided by known or estimated values of the materials). All phase retrieval parameters given in this table are necessary and sufficient for replication starting from the raw data, along with the procedures explained in the corresponding references. A detailed description of the different phase retrieval algorithms would exceed the framework of this manuscript.

l183: "corrupted datasets" What was the issues here? Need clarifications.

Some scans were affected by errors in the data transfer, which was noticed only after the available beamtime.

l197: "32 pixels for PB datasets" If the analysis has been done on binned data, it would lead to different size compare to the 12 pixels for the LJ acquisitions?"A smoothing parameter of 2 pixels" It would be nice to have more explanation here. Have this been applied to both PB and LJ datasets?

32 pixels refer to the unbinned data, as mentioned in the following sentence in the manuscript.

l213: "the paraffin surrounding" Why the mask has been applied only on the LJ datasets. Isn't it included as well in the acquisition of the PB datasets?

The statistical analysis based on the structure tensor has only be applied to the LJ data.

l214: "Since one axis of the shape measure is redundant" Could the author clarify this point?

The three shape measures add up to one, as described in line 210

l231: "A small number of axis-aligned 2D slices was annotated". It would be very interesting to know the amount of annotated datasets that have been necessary to perform the analysis. Indeed, this is the most critical point when performing machine learning technique. Sparsely annotated data sets is a very promising technique, specially for analysing large amount of images. What was the percentage of images of annotated volumes were kept for training / validation?

We did not keep exact account of the volume percentage where manual annotation was performed. In fact, manual segmentation of a single blood vessel can already suffice if the data quality is high and comparable between datasets.

l236: Why 96x96x96 voxels? What was the total size of the datasets?

Training and augmentation of data requires selection of small subvolumes, for two reasons: (1.) The net learns that the large configuration and borders are not important, and (2.) On the order of 10 subvolumes can be treated efficiently in parallel on the graphics card.

l239 "256 subvolumes" How is this amount representative of the entire datasets?

The subvolumes are randomly chosen. The segmentation results were validated by visual inspection.

l243: "A separate model" why was it necessary? Doesn't it create bias in the results?

For the analysis of the Covid-19 sample, a separate model needed to be trained, since the structure of the vasculature differs a lot from the controls: as described in the Results section, the blood vessels of the Covid-19 patients are mostly filled with blood while control vessels were mostly abundant of blood. In order to improve the quality of the segmentation the network had to be retrained for said differences.

l247: "adding additional annotations" Do I understand correctly that the analysis is run a first time, then the authors look at the images and visually correct some of them and re-run the analysis?

Yes, correct. This procedure improves the quality of the segmentation.

l314: "different areas of the same heart" How those areas have been chosen. Why 3 samples per patients?

Areas were chosen by eye and comparison to neighboring histological sections. 3 tissue blocks per patient (control group) were provided.

l323: "dark stripes" In the figure it's easy to distinguish white stripes, but not the dark ones. Maybe small areas or markers would help to follow the description.

The dark stripes (collagen sheets) are marked in Figure 3.

l329 / Figure 3: How to make the difference between paraffin cracks and inclusion compare to other features?

This is distinguished based on gray values (electron density).

l337: "samples near an artery". How the areas have been chosen and what is the impact on the results presented?

Samples were chosen based on the histology, in order to increase tissue content in the punch. In some cases it simply turned out that a medium size vessel was present in the chosen volume.

l360: Appendix 1 Figure 2: would be nice to have arrows in the figure to understand what is erythrocytes and capillaries.

This figure is intended only to compare the data quality between synchrotron and in-house; we would prefer not to distract from this by any additional labels.

l368: "340x340x340um3": Why this size has been chosen and not the entire volume?

The FOV is limited by the effective pixelsize and the amount of detector pixels (which is limited by its manufacturing).

l371: "cytosol" would be nice to have this indicated in the figure.

The myofibrils are located in the cytosol (background); we prefer to only indicate the myofibril here.

l375: "nucleus" It seems that we can see only on in the figure. Do we see others in the 3D volume?

Each cardiomyocyte within the volume contains one nucleus. In the figure we present one exemplary region, the high resolution data is publicly available and can be further investigated. Indeed, many more nuclei can be identified!

l386: "speeding-up the measurement sequence" It would be interesting to indeed grasp the difficulties of such measurement and time necessary.

For future work we plan to update the instrument (waveguide optics, automated alignment, continuous data acquisition).

Table 4: mean shape mu_p and mu_s are of the same order (ctrl or diseased) for mu_l there is a difference between control versus diseased however the σ is pretty large (almost 30% in certain case). Similar remarks for the elliptical fit (54% for σ in the case of Control…). The Authors are aware and found an interesting way of representing the results (Figure 5A). However, can the authors comments on why the errors are so huge, and how can one conclude in term of differences according to those results? Because even on the graph, we see tendency but to make strong concluding statement is difficult. I would therefore modify part of the eluding sentence in this direction.Maybe could they compare with histology analysis conclude beforehand to help in the understanding.

The large intra-group variance reflects the pronounced variability between individual subjects and samples, which is in line with experience of conventional histology. We have added a comment on this.

l403: Authors are aware that their methods have some limitation. For instance "depend on tissue preservation and preparations".

Sample preparation is always an issue when investigating biological tissues and it is important to take the sample quality into account.

l419: Welch t-test: Would be good to have a reference an explanation sentence.

Since the Welch t-test is a concept which was first published in 1947 and used as gold standard for statistical investigations we refrained from adding a reference.

l427: "Surface rendering" from which acquisition? Would be nice also to refer to the corresponding method paragraph.

We included the information on the corresponding datasets in the text and on the software used for visualization (avizo) in the methods section.

l438: Could the authors explains a bit more what is the probability density function and how it is obtained. Figure 6E: The authors state clearly that their is a higher amount of branching points in the Covid19 sample. However, this is the results on one sample and the figure is not clearly stating that. For instance for 5 vertex degree the control seems superior than the covid.

The PDF quantifies the node connectivity after skeletonization procedure, which is carried out according to ref. [Lee et al., 1994]. The PDF is normalized, all red (or green) columns add up to one. Now the different heights are a ‘fingerprint’ of the graph topology. The 5 vertex degree is negligible (extremely rare), and even 4 is marginal. This mainly serves to show that vertex degrees 1-3 are relevant here. This is now explained more clearly.

l449: "first report" What about Walsch 2021?

We present the first report comparing cardiac histopathology of Covid-19 to influenza and coxsackie myocarditis; we now state more clearly that Walsh et al., have already studied heart tissue in Covid-19, albeit with different sample preparation and at different resolution.

l465: "non-destructively". It's not totally non-destructive technique as one still need to make a biopsy punch on the blocks.

We removed non-destructively in this context. In general X-ray tomography is a destruction-free technique and the punches can be placed back into the blocks and/or be used for further investigations.

l470: "conventional histopathology assessment" It would be nice to have a reference on the histology conventional analysis are adding in supplementary material correlation between X-ray images and histology images.

We have made this important point in our previous manuscript on lung damage in Covid-19, and include this reference now in the methods section.

Figure 5: how those 2 samples have been chosen compare to the previous selection.

The samples in Figure 6 were chosen randomly.

l493: "volume throughput of 10ˆ7 um3 /s. Maybe a simpler presentation would be more meaningful for the general audience, like acquisition time for one sample with which setup.

Acquisition time per sample is now also given in the manuscript.

l497: rather than by photon flux: This is valid for the synchrotron acquisition I guess. But then what would be the purpose of the new synchrotron source if the flux is already not fully exploited? Same question concerning the following sentence about the attenuators used to prevent detector saturation. What about dose on the sample then?

This is true for PB acquisition, but not the WG configuration. This is now stated more precisely.

l500: why so huge acquisition time range 200ms to 2500ms?

The photon flux depends on the diameter of the waveguide channel, thus the acquisition time has to be adapted to achieve similar fluence.

Appendix 1 Figure 3: In all the images we see like a cross in the images creating a white blur. Can the authors comment on that? Specially because grey levels are used for the analysis. How this effect can affect the results? A comment should also be added for the square missing (i.e. areas for corrupted files). Why measurements have not be redone locally?

The intensity changes result from local tomography artefacts and appear in the stitched datasets. Since the structural analysis works not on the gray value but on its gradients, the effect can be neglected in the shape measure analysis. The information of corrupted datasets was included in the caption.

Reviewer #2 (Recommendations for the authors):It would be important to improve the samples' statistics.

In this work, we investigated 26 cardiac tissue samples from various diseases (Covid-19, coxsackie myocarditis and H1N1/A influenza) at the laboratory setup. More than 500GB raw data and for each volume approximately 2000 voxels cubed were analyzed in terms of structural shape measure characterization. In follow up studies, we plan to extend the sample statistics and use extensive mathematical models based on optimal transport to refine structural differences between individual sample groups. However, this relies on the availability of autopsy samples, which are always quite limited, since autopsies are not routinely performed of patients who succumbed to Covid-19.